# Mass spectrometry-based absolute quantification of amyloid proteins in pathology tissue specimens: Merits and limitations

**Makiko Ogawa**[1,2,3⊚], **Yukako Shintani-Domoto**[1⊚], **Yoshiki Nagashima**[4,5], **Koji L. Ode**[4], **Aya Sato**[6], **Yoshihiro Shimizu**[6], **Kenichi Ohashi**[7], **Michael H. A. Roehrl**[2,3], **Tetsuo Ushiku**[1], **Hiroki R. Ueda**[4,8], **Masashi Fukayama**[1,9]*

1 Department of Pathology, Graduate School of Medicine, The University of Tokyo, Tokyo, Japan,
2 Department of Pathology, Memorial Sloan Kettering Cancer Center, New York, New York, United States of America, 3 Human Oncology and Pathogenesis Program, Memorial Sloan Kettering Cancer Center, New York, New York, United States of America, 4 Department of Systems Pharmacology, Graduate School of Medicine, The University of Tokyo, Tokyo, Japan, 5 Thermo Fisher Scientific K.K., Yokohama, Kanagawa, Japan, 6 Laboratory for Cell-Free Protein Synthesis, RIKEN Center for Biosystems Dynamics Research, Suita, Osaka, Japan, 7 Department of Pathology, Graduate School of Medicine, Yokohama City University, Yokohama, Japan, 8 Laboratory for Synthetic Biology, RIKEN Center for Biosystems Dynamics Research, Suita, Osaka, Japan, 9 Asahi TelePathology Center, Asahi General Hospital, Asahi-City, Chiba, Japan

⊚ These authors contributed equally to this work.
* mfukayama-tky@umin.org

**Data Availability Statement:** All relevant data are within the paper and its Supporting Information files. The raw data of SRM analysis were deposited

## Abstract

To clarify the significance of quantitative analyses of amyloid proteins in clinical practice and in research relating to systemic amyloidoses, we applied mass spectrometry–based quantification by isotope-labeled cell-free products (MS-QBIC) to formalin-fixed, paraffin-embedded (FFPE) tissues. The technique was applied to amyloid tissues collected by laser microdissection of Congo red-stained lesions of FFPE specimens. Twelve of 13 amyloid precursor proteins were successfully quantified, including serum amyloid A (SAA), transthyretin (TTR), immunoglobulin kappa light chain (IGK), immunoglobulin lambda light chain (IGL), beta-2-microglobulin (B2M), apolipoprotein (Apo) A1, Apo A4, Apo E, lysozyme, Apo A2, gelsolin, and fibrinogen alpha chain; leukocyte cell–derived chemotaxin-2 was not detected. The quantification of SAA, TTR, IGK, IGL, and B2M confirmed the responsible proteins, even when the immunohistochemical results were not decisive. Considerable amounts of Apo A1, Apo A4, and Apo E were deposited in parallel amounts with the responsible proteins. Quantification of amyloid protein by MS-QBIC is feasible and useful for the classification of and research on systemic amyloidoses.

## Introduction

Amyloidosis comprises a large group of diseases in which misfolding of extracellular proteins plays a fundamental role. Dynamic processes, occurring in parallel with or as an alternative to

to the PeptideAtlas SRM Experiment Library (PASSEL) (http://www.peptideatlas.org/PASS/PASS01558).

**Funding:** This work was supported by The Japan Society for the Promotion of Science (Grant No. 17K08717 to Y.S-D.). The funder had no role in study design, data collection and analysis, decision to publish, or preparation of the manuscript. H.R.U conducted a collaborative research project with Thermo Fisher Scientific Inc. Y.N. is an employee of Thermo Fisher Scientific, Inc. The company provided support in the form of salary for Y.N., and technical advice on the setup of mass spectrometers. However, the company did not have any additional role in the study design, data collection and analysis, decision to publish, or preparation of the manuscript. M.H.R. acknowledges NCI R21 CA231109, NCI R21 CA251992, the Farmer Foundation, a Cycle for Survival Equinox Innovation grant, and partial funding through the MSKCC NIH/NCI Cancer Center Support Grant P30 CA008748. None of these companies had any influence in support, design, execution, data analysis, or any other aspect of this study. The specific roles of these authors are articulated in the 'author contributions' section.

**Competing interests:** H.R.U conducted a collaborative research project with Thermo Fisher Scientific Inc. Y.N. is an employee of Thermo Fisher Scientific, Inc. The company provided support in the form of salary for Y.N., and technical advice on the setup of mass spectrometers. M.H.R. is a member of the Scientific Advisory Boards of Proscia, Trans-Hit, and Universal DX. There are no patents, products in development or marketed products associated with this research to declare. This does not alter our adherence to PLOS ONE policies on sharing data and materials.

physiologic folding, generate insoluble and toxic protein aggregates that are deposited in tissues in β-sheet fibrils. More than 30 different proteins are known to cause amyloidosis [1]. The various forms of amyloidosis require different treatment approaches, including high-dose melphalan with stem cell transplantation in AL amyloidosis, and liver transplantation or use of transthyretin (TTR)-tetramer stabilizer in hereditary TTR amyloidosis [2, 3]. Treatments for systemic amyloidoses have advanced over the years, and precise typing is key for adequate treatment.

The diagnosis of amyloidosis requires identification of amyloid in tissue specimens. Congo red staining in conjunction with polarized light microscopy is the primary procedure for histological confirmation of amyloid deposits, which is followed by subtype classification. The approach for the subtyping of amyloidosis has evolved over the last 30 years, moving from histochemistry to more specific immunohistochemistry of the major precursor proteins, such as serum amyloid A (SAA), transthyretin (TTR), immunoglobulin kappa light chain (IGK), immunoglobulin lambda light chain (IGL), and beta-2-microglobulin (B2M). However, immunohistochemistry has an inherent weakness that often results in nonspecific staining that can lead to false positive results. The assessment of the immunostaining intensity of amyloid proteins is subjective or at most semiquantitative. More recently, liquid chromatography (LC)–tandem mass spectrometry (MS/MS) has been introduced for direct identification of the amyloid-forming protein in biopsy specimens in several special institutions throughout the world [4]. The LC-MS/MS method frequently detects multiple amyloid proteins (such as apolipoproteins), even in the presence of a single major dominant amyloid protein. This occurrence sometimes makes it difficult to determine the amyloid subtype [5–7]. These additional proteins may act as cofactors in amyloid deposition, but detailed quantitative analysis of these proteins has not yet been undertaken.

In the present study, we developed MS-based absolute quantification of amyloid proteins for the subtyping of systemic amyloidoses in formalin-fixed, paraffin-embedded (FFPE) tissue sections. We adopted the technique, MS-based quantification by isotope-labeled cell-free products (MS-QBIC), which was reported by Narumi et al [8]. The method was originally developed to quantify subtle variations in mouse circadian clock proteins. In the MS-QBIC method, isotope-labeled peptides are prepared for the internal control of LC-MS analysis by using a cell-free synthesis system. Therefore, we expected that the application of MS-QBIC to the diagnosis of amyloidosis would be feasible if we could design appropriate peptide sequences for reference quantification. Furthermore, quantification of the deposited amyloid proteins may be useful to understand possible interactive processes in the maturation of amyloid deposits in the tissue.

The present study is an attempt to answer the following three questions: (1) whether the MS-QBIC method worked for FFPE specimens; (2) whether the method was feasible (by comparing the quantitative data obtained by the MS-QBIC process with immunohistochemical data, which are used routinely in the subclassification of amyloidosis); and (3) how the deposited amounts of major amyloid proteins were compared with other amyloid proteins. Thus, we aimed to clarify the merits and limitations of a quantitative analysis of amyloid proteins in both clinical practice and in research on systemic amyloidosis.

## Materials and methods

### Patients and tissue samples

Specimens from 30 patients with systemic amyloidosis were retrieved from the files of autopsy records of the Department of Pathology of the University of Tokyo from January 1999 through December 2016. The diagnosis of systemic amyloidosis was based on the presence of amyloid

in FFPE tissue sections from more than two organs by Congo red staining. Patients' medical records were reviewed for the presence of a monoclonal gammopathy using immunofixation of serum and urine as well as a serum-free light-chain assay.

Amyloid deposition was evaluated using Congo red-stained tissue sections. The staining was confirmed in representative sections under a fluorescein isothiocyanate (FITC) filter on a BZ-X710 all-in-one fluorescence microscope (Keyence Ltd., Osaka, Japan). We selected two specimens from each patient for the study: one from the cardiac wall of the left ventricle (n = 30) and one each from another organ (n = 28). Only myocardial tissue was available for examination in two cases, since the amounts of amyloid were too small for the subsequent MS analyses. The amount of amyloid deposition was scored in five grades from minimal (grade 1) to severe (grade 5) in myocardial tissue sections (S1 Fig).

For the application of the MS-QBIC method to the absolute quantification of amyloid protein, target peptides need to be selected as an isotope-labeled internal control of for the MS analysis. The target peptides were determined by the rules as described below with reference to previous LC-MS analyses, which examined the specimens derived from patients with systemic amyloidosis and related multiple myeloma, and from those with local amyloidosis including diabetes mellitus and Alzheimer's disease (S1 Table). The materials were chosen with reference to the files of autopsy or biopsy records of the Department of Pathology of the University of Tokyo from January 1994 through December 2015.

The study was approved by Research Ethics Committee, Graduate School of Medicine, The University of Tokyo (No. 10461-4-(5)). Informed consent for the use of patient specimens for the research was obtained from the responsible parties.

## Immunohistochemistry

For the identification of amyloid protein, antibodies against four amyloid precursor proteins (SAA, TTR, IGK, B2M) were used: anti-SAA antibody (mouse monoclonal, mc1; Dako Denmark A/S, Glostrup, Denmark, dilution 1:1000), anti-TTR antibody (rabbit monoclonal, EPR3219; Abcam plc, Cambridge, UK, 1:200), anti-IGK antibody (rabbit monoclonal, H16-E; Abnova Corp., Taipei, Taiwan, 1:1000), and anti-B2M antibody (rabbit polyclonal; Dako, 1:1000) [9–16]. For IGL, anti-lambda (118–134) antiserum was provided by Prof. Yoshinobu Hoshii (Department of Laboratory Science, Yamaguchi University Graduate School of Medicine, Ube, Japan, 1:800) [13, 17]. Immunostaining was performed using the BenchMark ULTRA immunohistochemistry/in situ hybridization system (Roche Diagnostics, Basel, Switzerland). For the immunostaining of IGK and IGL, pretreatment was performed in formic acid for 5 min. In this study, the results of immunostaining were evaluated on a scale of 0 to 3+ by the immunostaining intensity and stained area as follows: 0 = negative, 1+ = weakly and focally positive, 2+ = intermediate in both intensity and area, and 3+ = strongly and diffusely positive.

## Construction of MS-QBIC-based methods for the quantification of amyloid proteins

Absolute quantification of amyloid proteins was performed according to the MS-QBIC workflow [8] (S2 Fig). MS-QBIC based quantification consists of following processes: selection of target peptide candidates, synthesis of the MS-QBIC peptides, quantification of the MS-QBIC peptides, confirmation of signal linearity and determination of the Lower Limit of quantification (LLOQ).

**Selection of target peptide candidates of amyloid proteins.** Thirteen amyloid precursor proteins were selected as target proteins on the basis of the guidelines on the management of

amyloidosis, 2010 [18] for evaluation: apolipoprotein A-1 (Apo A1), apolipoprotein A-2 (Apo A2), apolipoprotein A-4 (Apo A4), apolipoprotein E (Apo E), gelsolin, fibrinogen alpha chain (FGA), lysozyme, and leukocyte cell–derived chemotaxin-2 (LECT2), in addition to SAA, TTR, IGK, IGL, and B2M. To select the target peptides for MS-QBIC based quantification, several tissue samples (S1 Table) were analyzed by a data-dependent MS/MS analysis.

All the samples were digested to recover the peptides according to the phase-transfer surfactant (PTS) method [8, 19] with several modifications. In brief, sample was dissolved in PTS buffer [8, 19] and heated at 95˚C for 3 h to reverse the formalin-crosslinks. The sample was reduced with 100 mM dithiothreitol (FUJIFILM Wako Pure Chemical Corp., Tokyo, Japan) at room temperature for 30 min, and then alkylated with 1 M iodoacetamide (Sigma-Aldrich Co., St. Louis, MO, USA) at room temperature for 30 min. Next, the sample was digested by adding 1 μg of lysyl endopeptidase (Lys-C) (FUJIFILM Wako Pure Chemical Corp.). After incubating the samples at 37˚C for 3 h, 1 μg trypsin (Roche) was added and the mixture was further incubated at 37˚C overnight. The detergents were removed by ethyl acetate/trifluoroacetic acid (TFA) solution according to the PTS method. The digested samples were then desalted using a self-prepared C18 stage tip [20]. The peptides were solubilized in 2% acetonitrile and 0.1% TFA and loaded to the LC-MS system to be separated by a gradient using mobile phases A (0.1% formic acid/$H_2O$) and B (0.1% formic acid and 100% acetonitrile) at a flow rate of 300 nL/min (4% to 36% B in 20 min, 36% to 95% B in 1 min, 95% B for 5 min, 95% to 4% B in 1 min and 4% B for 18 min) with a home-made capillary column (length of 200 mm and inner diameter of 100 μm) packed with 3μm C18 resin (L-column2, Chemicals Evaluation and Research Institute, Japan). The eluted peptides were electrosprayed (1.8–2.3 kV) and introduced into the Q-Exactive MS instrument (Thermo Fisher Scientific, Tokyo, Japan) in positive ion mode with data-dependent MS/MS. The obtained raw data was subjected to database search (UniProt, reviewed human database as of May 7th 2014) with the Sequest HT algorithm implemented in Proteome Discoverer 1.4 (Thermo Fisher Scientific). The parameters for database searches were as follows. Peptide cleavage was set to trypsin. Missed cleavage was allowed up to 2 and minimum and maximum peptide length was 6–144 amino acids. The mass tolerances were set to 10 ppm for precursor ions and 0.02 Da for fragment ions. For modification conditions, carbamidomethylation at cysteine was set as fixed modification and oxidation at methionine was set as variable modification. A significance threshold of $P<0.05$ was applied.

The identified peptides derived from the amyloid precursor proteins were further selected according to the following rules: (i) the amino acid sequences were unique to the target protein, (ii) each peptide contained a lysine or arginine residue only at its C-terminus, (iii) peptides containing cysteine were permitted because we used a protocol to convert cysteine almost completely to its alkylated form by reduction and alkylation [8], (iv) peptides with natural variants reported in several publications (searched via UniProtKB, https://www.uniprot.org/) were excluded (references are listed in the S1 Appendix) except for TTR, SAA and immunoglobulin light chains because excluding all the reported variants for these peptides limits the number of available peptide sequences, (v) target peptides of immunoglobulin light chains (IGK and IGL) were selected via the amino acid residues in constant lesions, which were the antibody-binding sites, as detected via immunohistochemistry [13, 14, 17, 21–23]. For Apo AII and LECT2, the data-dependent MS/MS analysis only identified fewer than four kinds of peptides; so because we aimed to select at least four peptides per one amyloid precursor protein for the MS-QBIC method, peptide sequences that matched the above-mentioned criteria (i)-(v) were selected manually. The maximum length of manually selected peptides was set as25 amino acids. The selected target peptides are summarized in S2 Table.

**Synthesis of MS-QBIC peptides.** MS-QBIC peptides were prepared using the PURE system, which is a reconstituted cell-free protein synthesis system as described by Shimizu *et al.* [24]. Using the MS-QBIC vector [8] as a template, PCR was performed for the amplification of the templates of MS-QBIC peptide synthesis with a T7 promoter primer (`5'-GGGCCTAAT ACGACTCACTATAG-3'`) as a forward primer and appropriate reverse primers (S2 Table). PCR mixtures were directly added to yield 25 μL of PURE system reaction mixtures containing $^{13}C_6$ $^{15}N_4$ L-arginine and $^{13}C_6$ L-lysine (Thermo Fisher Scientific) as substitutes for non-labeled L-arginine and L-lysine, respectively. The mixtures were incubated at 37°C for 30 min, and then the synthesized peptides were purified using anti-FLAG M2 magnetic beads (Sigma-Aldrich Co., St. Louis, MO, USA), according to the manufacturer's instructions. The recovered peptides were then stored at −80°C until used.

**Quantification of MS-QBIC peptides.** The concentration of the MS-QBIC peptide (stable isotope-labeled peptide) was first determined by comparison with the ion peak intensity of commercially available bovine serum albumin (BSA) as follows. First, weighed BSA powder (≥98.0% purity, Sigma-Aldrich Co., St. Louis, MO, USA) was dissolved in PTS buffer. The MS-QBIC product was then mixed with BSA and the mixture was subjected to trypsin digestion according to the PTS method as described above except 100 mM tris (2-carboxyethyl) phosphine Hydrochloride (Nacalai Tesque Inc., Kyoto, Japan) was used instead of 100 mM dithiothreitol. The pre-heating procedure was not applied to digest the non-fixed BSA/MS-QBIC samples. To reduce the non-specific binding of peptides to sample tubes, all the processes were carried out in the presence of 1 pmol/μL of trypsin-digested α-enolase (Sigma-Aldrich Co., St. Louis, MO, USA) prepared according to the PTS method. The signal linearity of the quantification tag (a peptide sequence derived from BSA: S2 Fig) was confirmed by analyzing different amounts of BSA (S3A Fig), although the stable-isotope based quantification does not heavily depend on the linearity of the signal with an increase in the peptide amount [25].

For the quantification of the tag sequence and all MS-QBIC-based quantification, it was carried out by selected reaction monitoring (SRM) analysis using a TSQ Quantiva triple-stage quadrupole mass spectrometer (Thermo Fisher Scientific). The following parameters were selected: positive mode, scan width of 0.002 *m/z*, Q1 and Q3 resolutions of 0.7 full width of half maximum (FWHM), cycle time of 3 s, and gas pressure of 1.8 Torr. The spectrometer was equipped with an UltiMate 3000 RSLCnano nano-high performance liquid chromatography (HPLC) system (Thermo Fisher Scientific), and a PepMap HPLC trap column (C18, 5 μm, 100 Å; Thermo Fisher Scientific) for loading samples. Analytical samples were solubilized in 2% acetonitrile and 0.1% TFA, and separated by reversed-phase chromatography using a PepMap rapid separation liquid chromatography (RSLC) EASY-Spray column (C18, 3 μm, 100 Å, 75 μm × 15 cm; Thermo Fisher Scientific) using mobile phases A (0.1% formic acid/$H_2O$) and B (0.1% formic acid and 100% acetonitrile) at a flow rate of 300 nL/min (4% B for 5 min, 4% to 36% B in 20 min, 36% to 95% B in 1 min, 95% B for 5 min, 95% to 4% B in 1 min and 4% B for 13 min). The eluted material was directly electro-sprayed into the MS. SRM transitions of the target peptides were optimized using Pinpoint software, version 1.3 (Thermo Fisher Scientific) (S2 Table). The Quan Browser of the Xcalibur data system, version 2.2 (Thermo Fisher Scientific) was used for data processing. The calibration curve of the BSA-derived quantification tag (S3A Fig) shows linear increase of signal intensity according to the increase of injected BSA; thus the amount of each MS-QBIC peptide was estimated to be proportional to the quantified heavy (MS-QBIC product) to light (weighed BSA) ratio. To determine the stock concentration of the MS-QBIC peptide, 0.2 μL of stock solution containing 100 fmol of digested BSA was analyzed. All the quantifications were within the range of 0.1–10 heavy (quantification tag derived from MS-QBIC peptide) to light (the tag derived from digested BSA) ratios (S3B Fig).

**Confirmation of signal linearity for each MS-QBIC product.** The signal linearity for each MS-QBIC product was then evaluated by analyzing different amounts of MS-QBIC peptides (S2 Appendix). For the linear regression evaluation, we excluded MS-QBIC peptides that failed to have the linear regression line that could be fitted to ≥4 consecutive amounts with $R^2 > 0.95$ and a %difference for each amount <30. The MS-QBIC peptides were further investigated to determine if the peptides could be detected robustly in the presence of peptides recovered from tissue samples. Several test tissue samples were prepared and mixed with MS-QBIC peptides according to the method described in the next section. A few MS-QBIC peptides that could not be detected robustly in the presence of tissue samples were excluded for the absolute quantification steps.

**Lower limit of quantification.** Next, based on the liner regression analysis, we defined the Lower Limit of Quantification (LLOQ) by determining the minimum amount of each MS-QBIC peptide for which the intensity was within a 30% difference of the linear regression line obtained in the each calibration plot; if all analyzed amounts were within the 30% difference, the minimum amount used to produce the liner regression line was set as the LLOQ. The values of the LLOQ are shown in S3 Table.

## Laser Capture Microdissection (LMD) and enzymatic digestion

A 10 μm FFPE section of each tissue sample was stained with Congo red. Positively stained areas were dissected using the PALM MicroBeam LMD system (Carl Zeiss Meditec AG, Tokyo, Japan). The samples were collected using PTS buffer as reported previously [8, 19]. An area of 0.07 to 0.27 mm$^2$ was dissected from each sample, with one exception (0.07 mm$^2$). Three samples were collected from each section, and a total of 174 amyloid samples (3 samples × 2 tissues × 28 patients and 3 samples × 1 tissue × 2 patients) were analyzed individually.

Enzymatic digestion of the 174 amyloid samples and MS-QBIC peptides was performed basically according to the PTS protocol as previously described above. The samples collected in the PTS buffer were heated at 95°C for 3 h to reverse formalin-crosslinks, prior to the reducing step with 100 mM dithiothreitol. A 0.0007–0.0027 mm$^3$ volume of collected amyloid sample may contain 0.7–2.7 μg of protein, assuming that the amount of protein is approximately 10% of the wet weight of the tissue sample. Thus, the addition of 1 μg of Lys-C/trypsin into each sample should be sufficient to digest all the amyloid proteins. The digested samples were desalted using a self-prepared C18 stage tip and dissolved in water containing 2% acetonitrile and 0.1% TFA. The MS-QBIC peptide mixture was added to the sample at this stage and then the sample was subjected to the MS analysis.

## Mass spectrometry for the MS-QBIC based absolute quantification

A TSQ Quantiva triple-stage quadrupole mass spectrometer (Thermo Fisher Scientific) was used for SRM analysis as described above. The target peptide derived from the amyloid sample was quantified by comparing the intensity of the nonlabeled target peptide in the sample with that of the MS-QBIC peptide of known concentration, which had been added to the sample prior to the measurement.

At least two or more target peptides were quantified per amyloid precursor protein. The quantitative value of each amyloid precursor protein was presented as the quantified value (pmol) of the protein divided by the volume (mm$^3$) that was resected from each sample via LMD. We used the average of the quantitative value of the amyloid protein for three samples of the same specimen for further analyses. S3 Table summarizes the raw quantification values. Approximately 80% of the quantified amount was calculated within the range of 0.1–10 heavy (MS-QBIC peptide) to light (endogenous amyloid peptide) ratios, indicating that in most

cases, the concentrations of the spiked peptide and the quantification targets were well balanced. The remaining ~20% of quantification values were calculated within the range of 0.01–100 heavy to light ratios.

The chromatogram of precursor ion and the spectrum of fragments of ions for a representative peptide for all the quantified amyloid proteins (excluding LECT2 that we could not detect in this study) are presented in S3 Appendix. The chromatograms of the endogenous and isotope-labeled peptides indicated that the two co-eluted peaks (endogenous and MS-QBIC peptides) were clearly detected and almost no-overlapping with other peaks from the background occurred. This peak shape provided information regarding the purity of the MS-QBIC peptides, indicating that possible contaminants or unexpected sub-products of the MS-QBIC peptides made a negligible contribution to the quantification of selective ion monitoring using the triple quadrupole mass spectrometer. The raw data from the SRM analysis were deposited to the Peptide Atlas SRM Experiment Library (PASSEL) (http://www.peptideatlas.org/PASS/PASS01558; see S3 Table for the relationship between raw file names and quantified results).

## Amyloid precursor protein in nonamyloid samples

To evaluate the quantity of amyloid precursor protein in the background tissue (nonamyloid sample), their relative amounts to amyloid deposition were determined by comparing nonamyloid tissue samples with those of amyloid deposition. Nonamyloid and amyloid samples were differently labeled with light $CH_2O$ and heavy $CD_2O$ isotopes, respectively, through dimethyl labeling with formaldehyde [26].

The stromal components of 11 specimens devoid of amyloid (derived from seven patients with conditions unrelated to amyloidosis) were micro-dissected using LMD. Each microdissection contained an area of 2.0–2.8 $mm^2$ in specimens without amyloidosis, and three samples from each specimen were analyzed. The samples of amyloid deposition were similarly obtained from 10 amyloidosis specimens (three samples per specimen). A total of 33 nonamyloid samples and 30 amyloid samples were analyzed. Enzymatic digestion of amyloid samples and nonamyloid samples was performed by the PTS method as described above. The samples collected in the PTS buffer were heated at 95 ˚C for 3 h to reverse formalin-crosslinks, prior to the reducing step with 100 mM dithiothreitol. The trypsin-digested samples were desalted by a C18 stage tip as described above, and before the elution from the tip, dimethyl-labeling was applied to peptides trapped on the tip [27]. $CD_2O$ and $NaBH_3CN$ (heavy label) were added to the amyloid samples, while $CH_2O$ and $NaBH_3CN$ (light label) were added to the nonamyloid samples. After elution from the C18 stage tips, all 30 amyloid samples were mixed as a reference. The mixture of the amyloid samples was dispensed to the 33 nonamyloid samples.

The samples were analyzed by the Q-Exactive MS instrument as described in the "Selection of target peptide candidates of amyloid proteins" section, except for the gradient conditions of mobile phases A and B (4%–36% B in 55 min, 36%–95% B in 1min, 95%B for 5min, 95%–4% B in 1 min and 4% B for 8 min). The Sequest HT/Proteome Discoverer 1.4 parameters for the database searches were same as described above, except that dimethylation (heavy- and light-labeled form) at the peptide N-terminus and at lysine was set as a fixed modification. The heavy to light ratios of identified labeled peptides were quantified by Proteome Discoverer 1.4 (Thermo Fisher Scientific).

## Statistical analysis

Statistical analysis was performed using JMP software, version 5.0.1J (SAS Institute, Inc., Cary, NC, USA). To assess correlations between the four main causative amyloid proteins (SAA,

TTR, IGK, and IGL) and Apo A1, Apo A4, Apo E, and lysozyme, we calculated the Spearman's rank correlation coefficient.

## Results

### Immunohistochemical subclassification of systemic amyloidosis

Clinical information and involved organs used in the study are presented in Table 1. These cases are ordered considering the results of immunohistochemistry (Figs 1 and 2). In 24 of the 30 patients, amyloidosis was subclassified by clinical and pathological examination at autopsy. Seven patients had AA amyloidosis. Of seven patients with ATTR/TTR amyloidosis, two

Table 1. Demographic features of patients with systemic amyloidosis.

| Case No. | Age (y) | M/F | Amyloid subtype | Clinical diagnosis at autopsy | Main organ deposit used | Deposition grade in cardiac tissue |
|---|---|---|---|---|---|---|
| 1 | 59 | F | AA | Rheumatoid arthritis | Adrenal gland | 2 |
| 2 | 78 | F | AA | Granulomatosis with polyangiitis | Stomach | 1 |
| 3 | 71 | F | AA | Systemic lupus erythematosus, Sjögren syndrome | Submandibular gland | 5 |
| 4 | 48 | M | AA | Dilated cardiomyopathy | Thyroid | 3 |
| 5 | 64 | M | AA | Rheumatoid arthritis | Kidney | 5 |
| 6 | 49 | F | AA | Rheumatoid arthritis | Thyroid | 5 |
| 7 | 75 | F | AA | Diffuse large B-cell lymphoma, Sjögren syndrome | Tongue | 1 |
| 8 | 77 | F | ATTR | S/O: Senile systemic amyloidosis | Esophagus | 5 |
| 9 | 65 | M | ATTR | Familial amyloidosis | Bladder | 5 |
| 10 | 75 | M | ATTR | Familial amyloidosis | Thyroid | 3 |
| 11 | 76 | M | ATTR | Alkaptonuria | Tongue | 3 |
| 12 | 85 | M | ATTR | Hepatocellular carcinoma, S/O: Senile systemic amyloidosis | Not available | 2 |
| 13 | 85 | M | ATTR | S/O: Senile systemic amyloidosis | Not available | 2 |
| 14 | 84 | F | ATTR | S/O: Senile systemic amyloidosis | Stomach | 3 |
| 15 | 77 | M | ND | Infective cerebral aneurysm, Systemic amyloidosis | Esophagus | 3 |
| 16 | 67 | M | ND | Hypertrophic cardiomyopathy, Systemic amyloidosis | Thyroid | 2 |
| 17 | 68 | M | ND | Multiple myeloma | Rectum | 2 |
| 18 | 54 | M | ALκ | Multiple myeloma | Cecum | 5 |
| 19 | 72 | F | ALκ | Multiple myeloma | Diaphragm | 1 |
| 20 | 67 | F | ALκ | Multiple myeloma | Stomach | 3 |
| 21 | 63 | M | ALκ | Multiple myeloma | Spinal dura | 1 |
| 22 | 76 | M | ND | Lymphoplasmacytic lymphoma, Diffuse large B-cell lymphoma | Lymph node | 1 |
| 23 | 82 | M | ND | Systemic amyloidosis | Thyroid | 3 |
| 24 | 74 | F | ALλ | Multiple myeloma | Tongue | 3 |
| 25 | 57 | M | ALλ | Multiple myeloma | Kidney | 2 |
| 26 | 72 | F | ALλ | Primary amyloidosis (serum IgA-λ, BJP-λ) | Spleen | 3 |
| 27 | 69 | M | ND | S/O: Primary amyloidosis (BJP+) | Kidney | 5 |
| 28 | 60 | M | ALλ | S/O: Primary amyloidosis (BJP+) | Tongue | 5 |
| 29 | 77 | M | ALλ | Systemic amyloidosis | Tongue | 4 |
| 30 | 62 | M | AB2M | S/O: Dialysis related amyloidosis | Prostate | 2 |

Amyloid proteins were determined by immunohistochemistry. AA, Amyloid A amyloidosis; ALκ, amyloid light chain (kappa) amyloidosis; ALλ, amyloid light chain (lambda) amyloidosis; ATTR, transthyretin amyloidosis; AB2M, beta-2-microglobulin amyloidosis; DRA, dialysis-related amyloidosis; ND, not determined; S/O, suspected of.

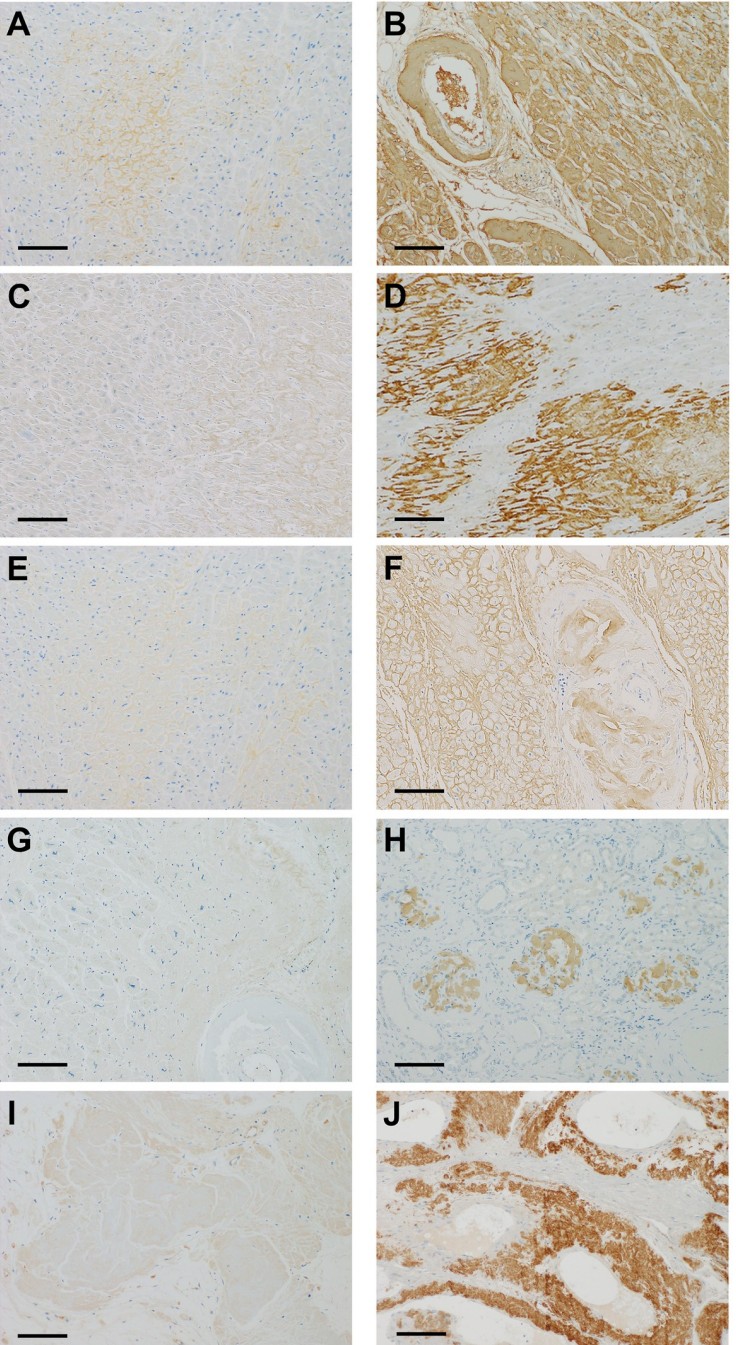

**Fig 1. Immunohistochemical staining of five major amyloid precursor proteins.** Immunostaining using 3,3′-diaminobenzidine. A and B: immunohistochemistry of serum amyloid A (SAA), showing weakly positive (1+) (A), and strongly positive (3+) (B). C and D: immunohistochemistry of transthyretin (TTR), showing weakly positive (1+) (C), and strongly positive (3+) (D). E and F: immunohistochemistry of immunoglobulin kappa light chain (IGK), showing weakly positive (1+) (E), and strongly positive (3+) (F). G and H: immunohistochemistry of immunoglobulin lambda light chain (IGL), showing weakly positive (1+) (G), and strongly positive (3+) (H). I and J: immunohistochemistry of beta-2-microglobulin (B2M), showing weakly positive (1+) (I), and strongly positive (3+) (J). Scale bars, 200µm.

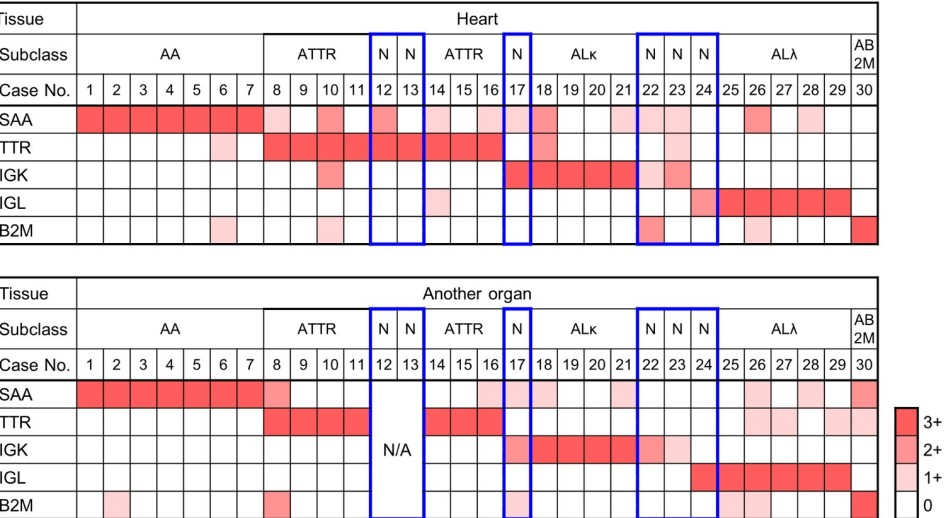

**Fig 2. Subclassification of amyloidosis by immunohistochemistry of major amyloid proteins.** The
Immunohistochemical results of serum amyloid A (SAA), transthyretin (TTR), immunoglobulin kappa light chain
(IGK), immunoglobulin lambda light chain (IGL), and beta-2-macroglobulin (B2M) were evaluated with a four-tiered
scales (0 to 3+). The scales are presented in white, and light, intermediate, and dark orange. Note the boxes delineated
in red show three sets of cases that did not meet the criteria for subclassification: (1) only one section available (#12
and #13), (2) no 3+ immunostaining (#22 and #23), and (3) 3+ and 2+ immunostaining in each pair (#17 and #24).
AA, AA amyloidosis; AB2M, beta-2-macroglobulin amyloidosis; ALκ, amyloid light chain (kappa) amyloidosis; ALλ,
amyloid light chain (lambda) amyloidosis; ATTR, ATTR amyloidosis; N, subclassification could not be determined at
autopsy; N/A, tissues not available.

patients had hereditary disease, and five patients had acquired disease. Six patients had associ-
ated multiple myeloma, and three patients were diagnosed as primary or systemic amyloidosis.
One patient had dialysis-associated amyloidosis.

Representative images of the immunohistochemical analysis using the panel antibodies are
presented in Fig 1. Strong homogeneous immunostaining (3+) of at least one protein was
observed in the entire amyloid deposits in 27 of 30 sections of the cardiac wall and in 25 of 28
sections of other organs (Fig 2). Positive results of 3+ were concordant in cardiac and other
organs in 24 patients, and there were no instances showing contradictory results in the paired
tissue sections. Relatively weak staining (2+) was accompanied by 3+ staining in four sections
from the cardiac wall (#10, 12, 18, and 26) and two sections from the other organs (#8 and 30).
Subclassification into five major types of amyloidosis was possible in 24 patients on the basis of
3+ immunostaining in the paired tissue sections, as shown in Table 1: AA amyloidosis, seven
patients (#1–7); ATTR amyloidosis, seven patients (#8-#11 and #14-#16); ALκ amyloidosis,
four patients (#18-#21); ALλ amyloidosis, five patients (#25-#29); and B2M amyloidosis, one
patient (#30).

There were three sets of specimens that did not meet the above criteria for the following
reasons: (1) there was only one section available (#12 and #13), (2) there was no 3+ immunos-
taining (#22 and #23), and (3) 3+ and 2+ immunostaining in each pair (#17 and #24).

## Quantification of amyloid proteins in LMD samples of Congo red–positive deposits

A total of 98 peptides were selected as the target peptides for 13 proteins (SAA, TTR, IGK,
IGL, B2M, Apo A1, Apo A2, Apo A4, Apo E, gelsolin, FGA, LECT2, and lysozyme). Fifty-three
peptides for 12 proteins were selected to obtain the absolute abundances of amyloid proteins

in LMD-dissected samples (Table 2). There were no peptides identified for quantification of LECT 2. The reasons for exclusion of peptides were as follows. (1) Two synthesized MS-QBIC peptides could not be detected by MS (potentially because of the low efficiency of the peptide synthesis by the PURE system or a low efficiency of ionization). (2) Twenty-six MS-QBIC

**Table 2. MS-QBIC peptides selected for the absolute quantification of amyloid proteins in LMD-dissected samples.**

| Protein(quantified/examined peptides) | Quantified peptide sequence | Non-quantified peptide sequence |
|---|---|---|
| SAA (2/4) | EANYIGSDK | GNYDAAK (2) |
| | DPNHFRPAGLPEK | AYSDMR (2) |
| TTR (2/7) | GSPAINVAVHVFR | ALGISPFHEHAEVVFTANDSGPR (2) |
| | AADDTWEPFASGK | YTIAALLSPYSYSTTAVVTNPK (2) |
| | | TSESGELHGLTTEEEFVEGIYK (2) |
| | | VEIDTK (2) |
| | | CPLMVK (2) |
| IGK (3/5) | TVAAPSVFIFPPSDEQLK | DSTYSLSSTLTLSK (2) |
| | SGTASVVCLLNNFYPR | VYACEVTHQGLSSPVTK (2) |
| | VDNALQSGNSQESVTEQDSK | |
| IGL (3/5) | AAPSVTLFPPSSEELQANK | ATLVCLISDFYPGAVTVAWK (2) |
| | AGVETTTPSK | SYSCQVTHEGSTVEK (2) |
| | YAASSYLSLTPEQWK | |
| B2M (3/6) | VNHVTLSQPK | SNFLNCYVSGFHPSDIEVDLLK (2) |
| | VEHSDLSFSK | DWSFYLLYYTEFTPTEK (2) |
| | IQVYSR | DEYACR (2) |
| Apo A1 (7/9) | VSFLSALEEYTK | AELQEGAR (2) |
| | QGLLPVLESFK | LAEYHAK (2) |
| | DLATVYVDVLK | |
| | EQLGPVTQEFWDNLEK | |
| | AKPALEDLR | |
| | LLDNWDSVTSTFSK | |
| | DYVSQFEGSALGK | |
| Apo A2 (2/4) | EQLTPLIK | EPCVESLVSQYFQTVTDYGK (4) |
| | SPELQAEAK | LLAATVLLLTICSLEGALVR (1) |
| Apo A4 (11/12) | SELTQQLNALFQDK | LNHQLEGLTFQMK (2) |
| | LAPLAEDVR | |
| | LVPFATELHER | |
| | LEPYADQLR | |
| | LLPHANEVSQK | |
| | LTPYADEFK | |
| | VNSFFSTFK | |
| | SLAPYAQDTQEK | |
| | ALVQQMEQLR | |
| | IDQNVEELK | |
| | IDQTVEELR | |
| Apo E (5/6) | AATVGSLAGQPLQER | ALMDETMK (4) |
| | LGPLVEQGR | |
| | FWDYLR | |
| | AQAWGER | |
| | QQTEWQSGQR | |

(*Continued*)

**Table 2.** (Continued)

| Protein(quantified/examined peptides) | Quantified peptide sequence | Non-quantified peptide sequence |
|---|---|---|
| Gelsolin (7/12) | EVQGFESATFLGYFK | HVVPNEVVVQR (4) |
| | TGAQELLR | FDLVPVPTNLYGDFFTGDAYVILK (2) |
| | QTQVSVLPEGGETPLFK | AQPVQVAEGSEPDGFWEALGGK (4) |
| | AGALNSNDAFVLK | VPVDPATYGQFYGGDSYIILYNYR (2) |
| | AVEVLPK | VPEARPNSMVVEHPEFLK (2) |
| | TPITVVK | |
| | EPGLQIWR | |
| FGA (6/17) | GLIDEVNQDFTNR | DSHSLTTNIMEILR (2) |
| | QLEQVIAK | MELERPGGNEITR (2) |
| | NSLFEYQK | HPDEAAFFDTASTGK (4) |
| | GGSTSYGTGSETESPR | VQHIQLLQK (4) |
| | AQLVDMK | ESSSHHPGIAEFPSR (4) |
| | GSESGIFTNTK | MKPVPDLVPGNFK (4) |
| | | QHLPLIK (4) |
| | | LEVDIDIK (2) |
| | | ALTDMPQMR (2) |
| | | TVIGPDGHK (4) |
| | | VSEDLR (3) |
| LECT2 (0/5) | | LGTLLPLQK (4) |
| | | SSNEIR (4) |
| | | NAINNGVR (3) |
| | | HGCGQYSAQR (2) |
| | | MFYIKPIK (1) |
| Lysozyme (2/6) | STDYGIFQINSR | TPGAVNACHLSCSALLQDNIADAVACAK (4) |
| | AWVAWR | GISLANWMCLAK (2) |
| | | LGMDGYR (4) |
| | | WESGYNTR (4) |

(1) The synthesized MS-QBIC peptide cannot be detected by mass spectrometers, potentially due to the low efficiency of the peptide synthesis by the PURE system or low efficiency of ionization. (2) These MS-QBIC peptides were not used for the absolute quantification because of the low quality of signal linearity evaluation. (3) These MS-QBIC peptides were not used for the absolute quantification because the peptides were not detected robustly when they were mixed with LMD samples. (4) These peptides derived from the LMD samples could not be separated from other adjacent signals or could not be detected in the samples. Apo A1, apolipoprotein A-1; Apo A2, apolipoprotein A-2; Apo A4, apolipoprotein A-4; Apo E, apolipoprotein E; B2M, beta-2-microglobulin; FGA, fibrinogen alpha chain; IGK, immunoglobulin kappa light chain; IGL, immunoglobulin lambda light chain; LRCT2, leukocyte cell-derived chemotaxin-2; MS, mass spectrometry; MS-QBIC, mass spectrometry–quantification by isotope-labeled cell-free products; SAA, serum amyloid A; TTR, transthyretin.

peptides were excluded because of the low quality of the linear regression lines. (3) Two MS-QBIC peptides were also excluded for the absolute quantification because these peptides were not quantified robustly when mixed with peptides derived from LMD samples. (4) Fifteen peptides had signals that could not be detected or separated from the adjacent signals in the LMD sample.

Three samples from each specimen were subjected to analysis of 12 proteins. The average of the quantitative value of the peptides for each protein was used for further analyses, and the results of the quantification are shown in Table 3. There were three groups of amyloid proteins in the present study: Group 1, amyloid proteins with immunohistochemical analyses of deposition (SAA, TTR, IGK, IGL, B2M); Group 2, certain amounts of proteins detected without

**Table 3. Quantitative values of the amyloid proteins.**

| Amyloid protein | Range (pmol/mm³) | Median (pmol/mm³) | Mean ± SD (pmol/mm³) | Immuno-histochemistry |
|---|---|---|---|---|
| SAA | 0.000–268.439 | 0.770 | 33.577 ± 69.440 | yes |
| TTR | 0.000–367.852 | 0.854 | 63.902 ± 112.086 | yes |
| IGK | 0.000–128.572 | 3.857 | 17.539 ± 31.021 | yes |
| IGL | 0.000–108.994 | 3.274 | 10.138 ± 18.590 | yes |
| B2M | 0.307–77.763 | 0.638 | 2.973 ± 12.185 | yes |
| Apo A1 | 0.000–21.918 | 2.731 | 3.951 ± 4.186 | no |
| Apo A2 | 0.000–0.967 | 0.000 | 0.074 ± 0.224 | no |
| Apo A4 | 0.523–36.247 | 5.933 | 7.103 ± 6.602 | no |
| Apo E | 1.543–46.008 | 9.637 | 13.074 ± 9.767 | no |
| Gelsolin | 0.098–1.766 | 0.595 | 0.644 ± 0.374 | no |
| FGA | 0.000–4.968 | 1.221 | 1.385 ± 1.210 | no |
| Lysozyme | 2.104–23.802 | 4.477 | 4.950 ± 2.962 | no |

Apo A1, apolipoprotein A-1; Apo A2, apolipoprotein A-2; Apo A4, apolipoprotein A-4; Apo E, apolipoprotein E; B2M, beta-2-microglobulin; FGA, fibrinogen alpha chain; IGK, immunoglobulin kappa light chain; IGL, immunoglobulin lambda light chain; LRCT2, leukocyte cell-derived chemotaxin-2, SAA, serum amyloid A; TTR, transthyretin

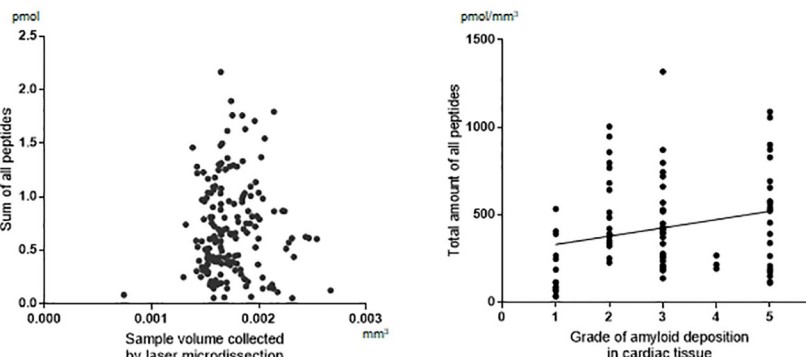

**Fig 3. Recovery of amyloid proteins from tissue specimens.** A. Effect of tissue collection by laser microdissection. Scatter plot of the total amounts of peptides (pmol) in mass spectrometry (MS) quantification and the area collected by laser microdissection. B. Effect of the amount of tissue amyloid. Distribution plot of the density of amyloid protein peptides (pmol/mm³) by the deposition grades of the cardiac tissues. Spearman's rank correlation, R = 0.2707, P = 0.0099.

evidence of specific deposition (Apo A1, Apo A4, Apo E, and lysozyme); and Group 3, extremely low amounts detected (Apo A2, gelsolin, and FGA).

For the effect of LMD procedures on the recovery of amyloid proteins from the tissue, there was no significant correlation between the total amount of peptides in MS-quantification and the area collected by LMD (Fig 3A). The amyloid deposition grades showed positive correlation with increased total peptides, although the correlation was relatively weak (R = 0.2707) (Fig 3B).

## Type-specific amyloid protein deposition

When each protein amount was compared with the immunohistochemical results (Fig 4), all samples in the 3+ grade of immunohistochemistry showed higher amounts of the amyloid

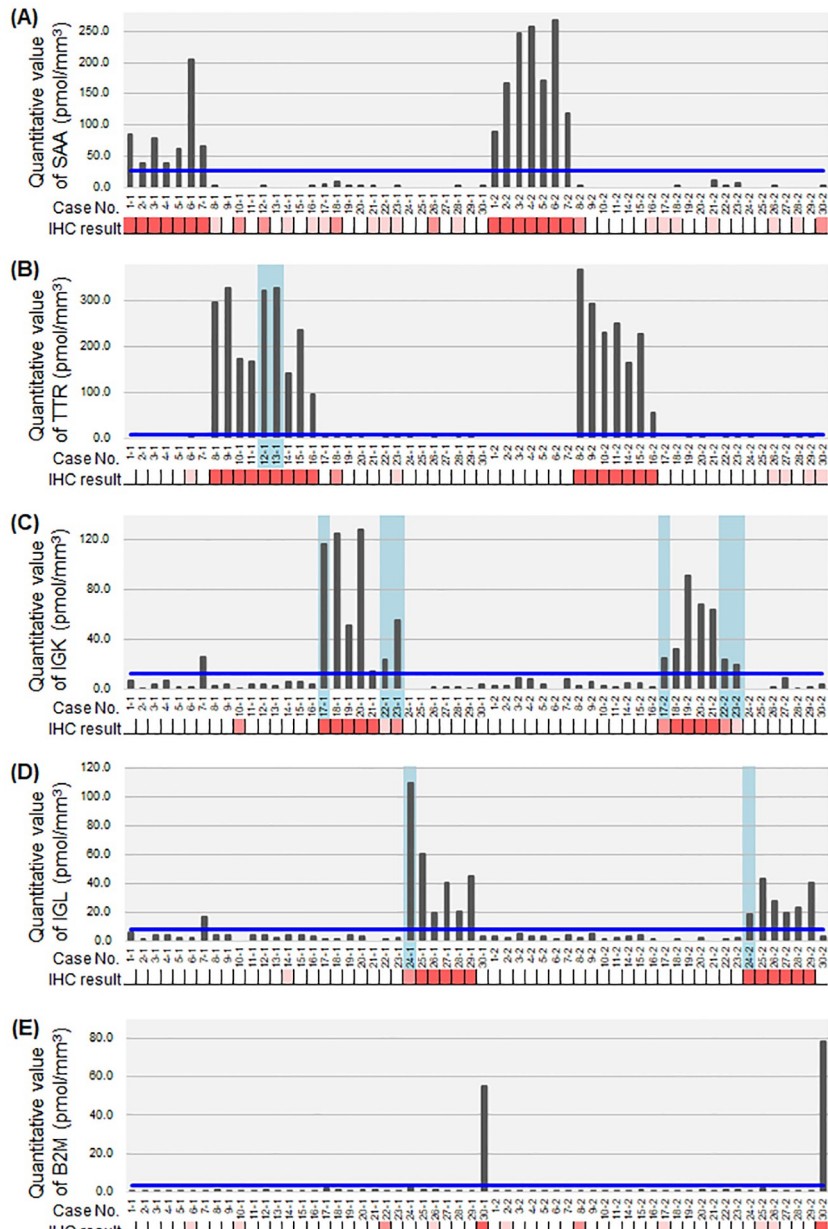

**Fig 4. Mass spectrometry quantification of major amyloid proteins with corresponding immunohistochemical scores.** Mass spectrometry (MS) quantification of serum amyloid A (SAA) (A), transthyretin (TTR) (B), immunoglobulin kappa light chain (IGK) (C), immunoglobulin lambda light chain (IGL) (D), and beta-2-microglobulin (B2M) (E). Immunohistochemical results of each corresponding protein are presented as the scales: 0, white; 1+, light orange; 2+, intermediate orange; and 3+, dark pink. The cases with an undetermined subclass are highlighted in a light blue-colored box. The blue line represents the proposed cut-off value for determination of subclass as shown in Table 4.

protein. In patients #12 and #13, in whom the immunohistochemical results were confirmed in only one section, the quantification provided further evidence of the diagnosis. In patients #17, #22, #23, and #24, the quantification confirmed the suspected diagnosis, even if immuno-histochemical results lacked enough evidence. There were two samples that showed higher than the lowest value in the corresponding subgroups, IGK and IGL in #7–1. The clinical

**Table 4. Proposed cut-off values for determination of the major amyloid subclasses.**

| Protein | Corresponding subgroup (pmol/mm³) | | Other subgroups (pmol/mm³) | | Cut-off value |
|---|---|---|---|---|---|
| | Mean ± SD | Range min-max | Mean ± SD | Max (#sample) | |
| SAA | AA subgroup (n = 14/7 cases) | | n = 34 | | max+6SD |
| | 134.976 ± 80.031 | 38.054–268.439 | 1.188 ± 2.466 | 11.484 (#21–2) | 26.280 |
| TTR | ATTR subgroup (n = 14/7case) | | n = 34 | | max+6SD |
| | 216.195 ± 85.117 | 56.514–367.852 | 0.538 ± 0.898 | 3.499 (#27–2) | 8.887 |
| IGK | ALk subgroup (n = 8/4cases) | | n = 40 | | mean+2SD |
| | 71.898 ± 38.314 | 14.751–128.572 | 4.214 ± 4.371 | 26.404 (#7–1) | 12.956 |
| IGL | ALl subgroup (n = 10/5cases) | | n = 38 | | mean+2SD |
| | 33.771 ± 13.212 | 19.037–60.228 | 2.912 ± 2.694 | 16.742 (#7–1) | 8.300 |
| B2M | AB2M subgroup (n = 2) | | n = 46 | | max+6SD |
| | - | 55.141–77.763 | 0.641 ± 0.303 | 1.837 (#25–2) | 3.655 |

Analysis was performed using the data of immunohistochemically determined subgroups (each corresponding amyloid subgroup vs. other subgroups)

B2M, beta-2-microglobulin; IGK, immunoglobulin kappa light chain; IGL, immunoglobulin lambda light chain; SAA, serum amyloid A; TTR, transthyretin

diagnosis in this patient was Sjögren syndrome complicated by lymphoma. The results suggest a small amount of light chain deposition in this specific case.

For setting the cut-off value for MS-quantification in the present study, we used a simulation as shown in Table 4. Based on the protein amounts in the immunohistochemically determined groups, the cut-off value for MS-quantification was at first set as the mean+2 SD of each protein. However, in SAA, TTR, and B2M, the difference between the minimum amount in each corresponding group was much higher than in the maximum of the other groups. Therefore, in these cases, the cut-off values were arbitrarily set as the maximum +6 SD.

## Non-type-specific protein deposition

Apo A1, Apo A4, Apo E, and lysozyme were also detected in all 58 specimens from 30 patients (Fig 5). Their absolute amounts were less than the minimum amounts of SAA and TTR. However, two samples each showed an excess amount of Apo A1 and Apo A4 relative to the minimum of ALκ. Many samples showed an excess amount of Apo E relative to the minimum of ALκ and ALλ. One sample showed an excess amount of lysozyme relative to the minimum of ALλ.

It is possible that the deposition of these proteins has some role in amyloid deposition; for example, as a facilitating cofactor. To test this assumption, we conducted additional analyses. First, to evaluate quantitative correlations between the four main type-specific amyloid proteins (SAA, TTR, IGK, and IGL) and Apo A1, Apo A4, Apo E, and lysozyme, we calculated the Spearman's rank correlation coefficient (Table 5), demonstrating that positive correlations were present except for lysozyme.

Second, the amounts of these proteins in nonamyloid tissues were measured relative to those in amyloid deposition (Table 6). The mean+2SD for these ratios was less than 0.5, except for lysozyme.

## Discussion

The MS-QBIC method enables high-throughput preparation of internal standards by using a reconstituted cell-free protein synthesis system and thereby facilitates multiplexed quantification of absolute amounts of target proteins with high sensitivity. The present study is the first

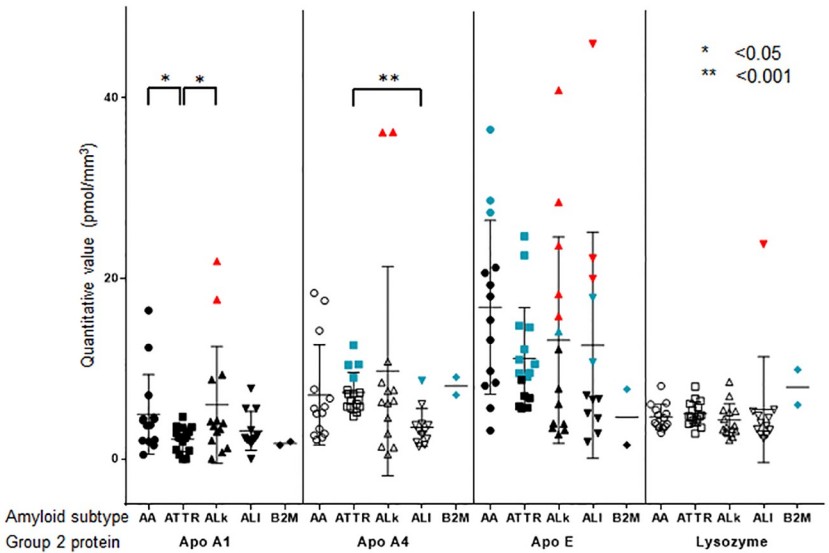

**Fig 5. Mass spectrometry quantification of Group 2 proteins in each of five major amyloid subtypes.** Bee swarm plot of the amount of Group 2 proteins (apolipoprotein A-1, Apo A1; apolipoprotein A-4, Apo A4; apolipoprotein E, Apo E; and lysozyme) by five major amyloid subtypes (serum amyloid A, SAA, circle; transthyretin, TTR, rectangle; immunoglobulin kappa light chain, IGK, triangle; immunoglobulin lambda light chain, IGL, inverted triangle; and beta-2-microglobulin, B2M, rhombus). The samples with the values higher than the arbitrarily set cut-off value for each corresponding protein, are labeled in blue-green. The samples with values higher than the minimum value for each corresponding protein, are labeled in red. Statistical differences in the protein amounts are presented when significant. MS, mass spectrometry.

**Table 5. Spearman's rank correlation coefficient for Group 2 amyloid proteins with Group 1 amyloid proteins in major amyloid subgroups.**

| Subgroup / Protein | | Apo A1 | | Apo A4 | | Apo E | | Lysozyme | |
|---|---|---|---|---|---|---|---|---|---|
| | | r | p | r | p | r | p | r | p |
| AA | SAA | 0.352 | NS | 0.414 | <0.01 | 0.367 | NS | 0.122 | NS |
| ATTR | TTR | 0.450 | <0.01 | 0.372 | <0.01 | 0.493 | <0.001 | -0.117 | NS |
| ALk | IGK | 0.575 | <0.001 | 0.508 | <0.01 | 0.694 | <0.0001 | 0.073 | NS |
| ALλ | IGL | 0.082 | NS | 0.661 | <0.0001 | 0.121 | NS | 0.265 | NS |

B2M, beta-2-microglobulin; IGK, immunoglobulin kappa light chain; IGL, immunoglobulin lambda light chain; SAA, serum amyloid A; TTR, transthyretin

**Table 6. Relative ratios of Group 2 amyloid proteins found in nonamyloid tissue.**

| Protein | Ratio Mean ± SD |
|---|---|
| Apo A1 | 0.137 ± 0.139 |
| Apo A4 | 0.099 ± 0.171 |
| Apo E | 0.013 ± 0.015 |
| Lysozyme | 0.321 ± 0.465 |

Apo A1, apolipoprotein A-1; Apo A2, apolipoprotein A-2; Apo A4, apolipoprotein A-4; Apo E, apolipoprotein E.

application of this method in the field of pathology. We chose the classification of systemic amyloidosis as the subject of this application because it is sometimes difficult to determine by immunohistochemistry and even by label free LC-MS/MS analysis.

Immunohistochemistry is the most common method used in pathology laboratories for subtyping amyloidosis. In the present study, using the panel of five antibodies against five major amyloid proteins, we encountered three kinds of difficulty: (1) there was only one section available, (2) there was no 3+ immunostaining, and (3) there was 3+ and 2+ immunostaining in each pair. Furthermore, there were many specimens that showed 2+ immunostaining in addition to 3+ immunostaining relative to other proteins. Absolute quantification of amyloid protein by MS-QBIC demonstrated strict correspondence between 3+ immunostaining and protein amounts. We observed false positive 2+ immunostaining in patients with AA (six of six specimens) and ATTR (one of one specimen), while a true positive was observed in patients with ALκ (three of four specimens) and ALλ (one of one specimen). With these results, we could strengthen the immunohistochemical results even in single section and clarify the significance of 2+ immunostaining for each antibody. The absolute quantification of amyloid protein could resolve all three problems of immunohistochemistry described above. Thus, the combination of more than 2+ immunostaining and absolute quantification of IGK and IGL could be the criteria to determine both subtypes.

There is a question as to whether Apo A1, Apo A4, Apo E, and lysozyme are nonspecifically deposited in the amyloid. The present study demonstrated differences between the three apolipoproteins and lysozyme. There was quantitative correlation between apolipoproteins (Apo A1, A4, and E) and each type-specific protein in the amyloid, but not in lysozyme. In nonamyloid background tissues, the apolipoproteins deposited were less than 0.5 (at most) compared with those in the amyloid deposits. Apo E and Apo A4 sometimes coexist with the main causative amyloid proteins [5–7, 28]. Amyloid deposits contain many kinds of biological molecules, such as apolipoprotein, serum amyloid P, and glycoprotein [29]. These molecules can constitute a scaffold, facilitating the initial phases of fibril nucleation, and they have a targeting role in the localization of amyloid deposits in the tissue. Apo E, Apo A4, serum amyloid P, and vitronectin are associated with amyloid fibrils and are involved in amyloid formation. Apo E is a common constituent of amyloid deposits in Alzheimer's disease. The major effect of Apo E isoforms occurs via the effect on Aβ aggregation and clearance, influencing the onset of Aβ deposition [28]. The Apo A4 fragment is strongly fibrillogenic *in vitro* and enhances fibril formation from wild-type TTR [5]. The apolipoproteins may facilitate the genesis of amyloid, especially in the cases of Apo A4 with ATTR and ALλ, and Apo E with ALκ. However, a characteristic of the lysozyme protein is a changeable conformation [30]. Enrichment of this protein may be passive and unrelated to the genesis of amyloid.

The amounts of Apo E and lysozyme proteins frequently exceeded the minimal amounts of IGK and IGL in ALκ and ALλ amyloidosis, respectively. It is possible that amyloidogenic immunoglobulin light chains, working as an initiator, precipitate other proteins in greater amounts than the initiator. Alternatively, these facts indicate a limitation of the method of absolute quantification in the present study. Many mutations in immunoglobulin light chains, either in the constant or the variable region, might cause complex conformation changes, resulting in difficulty in obtaining enough amounts of peptides from amyloid deposition of IGK and IGL.

In the present study, there was one patient with Sjögren syndrome and lymphoma, with deposition of SAA, IGK, and IGL but negative immunohistochemical results. The heavy mutation burden in immunoglobulin light chains might have prevented detection by immunohistochemistry. Alternatively, accumulated SAA protein covered the epitopes of immunoglobulin molecules. Although the LS-MS/MS method improves detection sensitivity by improving data

matching techniques, some peptides with mutations might also escape detection in LC-MS/MS analysis. In these circumstances, absolute quantification will identify such a case and contribute to the understanding of the complicated processes involved in amyloidosis.

Absolute quantification is potentially useful to compare the various effects of different types of amyloid in the diseased tissue. However, there are limitations in the present study. The effect of formalin fixation might not be uniform at each amino acid residue in a single protein, which may affect the accuracy of quantification given by a limited number of peptide candidates for one amyloid protein. Second, the amount of amyloid deposits was evaluated in units of pmol/mm$^3$ with the volume of amyloid as the denominator. This evaluation ignores the spatial distribution of amyloid proteins within the deposit area. However, the physical properties of each amyloid protein might be different if, for example, high concentrations of a particular type of amyloid protein are deposited in a particular region of the deposit area.

In conclusion, we successfully applied the MS-QBIC method to the absolute quantification of amyloid deposits in systemic amyloidosis. The quantitative data clarified the significance of immunohistochemical results and provided basis for the interpretation of amyloidosis classification by immunohistochemical panel analysis. Furthermore, the quantification by MS analysis disclosed the significance of apolipoproteins, which are components of amyloid aggregates. Absolute quantification of amyloid protein by MS-QBIC is a feasible and useful complement for the classification of and research into systemic amyloidosis.

## Supporting information

**S1 Appendix. References for the determination of peptide sequences.**
(PDF)

**S2 Appendix. Signal linearity of each MS-QBIC peptide measured by mass spectrometry.**
(PDF)

**S3 Appendix. Chromatogram of precursor ions and spectrum of fragment ions for a representative peptide of the quantified amyloid proteins.** Case ID (e.g., P1) and raw file ID (e.g., A003) correspond to the ID numbers shown in S3 Table. (A) and (M) serum amyloid A (SAA). (B) and (N) transthyretin (TTR), (C) and (O) immunoglobulin kappa light chain (IGK), (D) and (P) immunoglobulin lambda light chain (IGL), (E) and (Q) beta-2-microglobulin (B2M), (F) and (R) apolipoprotein (Apo) A1, (G) and (S) Apo A2, (H) and (T) Apo A4, (I) and (U) Apo E, (J) and (V) gelsolin, (K) and (W) fibrinogen alpha chain (FGA), (L) and (X) lysozyme, (A-L) specimens are derived from the heart, and (M-X) from other organs.
(PDF)

**S1 Fig. Amyloid deposition by Congo red staining.** The amount of Congo red-positive deposition was scored in five grades 1–5. from minimal to severe (A, C, E). The staining is well visualized under the FITC filter of a BZ-X710 all-in-one fluorescent microscope (B, D, F). Grade 2 (A and B), Grade 3 (C and D), and Grade 5 (E and F). Scale bars, 200μm.
(PDF)

**S2 Fig. Workflow of quantification of amyloid proteins by Mass Spectrometry–based Quantification By Isotope-labeled Cell-free products (MSQBIC).** The workflow has two parts: the synthesis and quantification of the MS-QBIC peptides (left side) and the quantification of target peptides using the MS-QBIC peptide as reference (right side).
(PDF)

**S3 Fig. Supporting information for the estimation of the concentration of MS-QBIC peptides.** (A) Signal linearity of the quantification tag measured by mass spectrometry. (B) Heavy

to light ratios of quantification tag signals used to estimate the concentrations of MS-QBIC peptides.
(PDF)

**S1 Table. Specimens used for data-dependent MS/MS analysis.**
(DOCX)

**S2 Table. List of MS-QBIC target peptides, primer sequences for the production of MS-QBIC peptides, and SRM method for the quantification.**
(XLSX)

**S3 Table. List of information used for the absolute quantification of amyloid proteins in the tissue samples, including the intensities and heavy/light ratios of MS-QBIC- and tissue-derived peptide signals.**
(XLSX)

## Acknowledgments

We thank Andrea Baird, MD, from Edanz Group (https://en-author-services.edanzgroup. com/) for editing a draft of this manuscript.

## Author Contributions

**Conceptualization:** Yukako Shintani-Domoto, Koji L. Ode, Kenichi Ohashi, Hiroki R. Ueda, Masashi Fukayama.

**Data curation:** Yoshiki Nagashima, Koji L. Ode.

**Funding acquisition:** Yukako Shintani-Domoto.

**Investigation:** Makiko Ogawa, Yukako Shintani-Domoto, Koji L. Ode, Aya Sato.

**Methodology:** Yoshiki Nagashima, Koji L. Ode, Aya Sato, Yoshihiro Shimizu.

**Project administration:** Masashi Fukayama.

**Resources:** Makiko Ogawa, Yukako Shintani-Domoto, Yoshiki Nagashima.

**Supervision:** Yoshihiro Shimizu, Michael H. A. Roehrl, Tetsuo Ushiku, Hiroki R. Ueda, Masashi Fukayama.

**Writing – original draft:** Makiko Ogawa, Koji L. Ode.

**Writing – review & editing:** Yukako Shintani-Domoto, Yoshiki Nagashima, Yoshihiro Shimizu, Kenichi Ohashi, Michael H. A. Roehrl, Tetsuo Ushiku, Hiroki R. Ueda, Masashi Fukayama.

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
