## [Decision Letter · Decision Letter 0]

19 Feb 2020

PONE-D-20-02311

Mass spectrometry–based absolute quantification of amyloid proteins in pathology tissue specimens: merits and limitations

PLOS ONE

Dear Prof. Fukayama,

Thank you for submitting your manuscript to PLOS ONE. After careful consideration, we feel that it has merit but does not fully meet PLOS ONE’s publication criteria as it currently stands. Therefore, we invite you to submit a revised version of the manuscript that addresses the points raised during the review process.

We would appreciate receiving your revised manuscript by Apr 04 2020 11:59PM. To enhance the reproducibility of your results, we recommend that if applicable you deposit your laboratory protocols in protocols.io, where a protocol can be assigned its own identifier (DOI) such that it can be cited independently in the future. For instructions see: http://journals.plos.org/plosone/s/submission-guidelines#loc-laboratory-protocols

We look forward to receiving your revised manuscript.

Kind regards,

Harald Mischak

Academic Editor

PLOS ONE

Journal Requirements:

2. Please include your tables as part of your main manuscript and remove the individual files. Please note that supplementary tables (should remain/ be uploaded) as separate "supporting information" files

"This work was supported by The Japan Society for the Promotion of Science (Grant No. 17K08717 to Y.S-D.).

The funder had no role in study design, data collection and analysis, decision to publish, or preparation of the manuscript."

We note that one or more of the authors are employed by a commercial company: 'Thermo Fisher Scientific K.K., Yokohama, Kanagawa, Japan'.

Additional Editor Comments (if provided):

As you can see, both authors found the paper potentially of value, but indicated multiple issues that need to be addressed. Many of these are required to ascertain validity of the results, and reproducibility. I agree with the comments, and hope that you can revise the manuscript accordingly

Reviewers' comments:

Reviewer's Responses to Questions

**Comments to the Author**

1. Is the manuscript technically sound, and do the data support the conclusions?

Reviewer #1: Yes

Reviewer #2: Partly

2. Has the statistical analysis been performed appropriately and rigorously? 

Reviewer #1: Yes

Reviewer #2: Yes

3. Have the authors made all data underlying the findings in their manuscript fully available?

Reviewer #1: Yes

Reviewer #2: No

4. Is the manuscript presented in an intelligible fashion and written in standard English?

Reviewer #1: Yes

Reviewer #2: Yes

5. Review Comments to the Author

Reviewer #1: Dear Authors and Editor,

The authors have applied their previously published MS-QBIC to quantitate the amyloids present in FFPE tissue samples. Their skill using mass spectrometry, pathology, and microdissection is obvious. They used PTS method to extract the proteins from the FFPE tissues. PTS is most appropriate for unfixed tissue.

Formalin fixing uses formaldehyde to crosslink proteins by their lysine and other residues. The authors used stable isotope-labeled formaldehyde in this work and they ensured that the consequent imine formation was reduced using sodium cyanoborohydride. The crosslinking resulting from formalin fixing would make some proteins inaccessible to trypsin and prevent some trypsin cleavage on proteins that were accessible to trypsin. The authors used the PTS method to extract proteins from this fixed tissue. This would assist in extracting the proteins, but it does not undo the formaldehyde mediated cross linking of the formalin. Other researchers use high temperatures and free amines to reverse the fixation process and yield intact proteins.

The authors need to revise the manuscript by including an additional limitation: namely that fixation may not be uniform for each protein. This may be why the authors were able to observe some tryptic peptides and not others. It also means that discussions about quantitation of proteins makes no sense when the authors cannot ensure that the proteins they are quantitating are extracted in proportion to their prevalence.

Some specific changes are listed below.

The authors use acronyms such as PTS (page 8) before they define it as phase transfer surfactants (page 10). They should carefully check the manuscript to ensure that the acronyms are defined before they are used.

The statement: “The various forms of amyloidosis require different approaches, including high-dose…”

Should be revised to include “treatment”:

“The various forms of amyloidosis require different treatment approaches, including high-dose…”

“13C6 15N4 L-arginine and 13C6 L-lysine.” Is Lysine unlabeled with 15N?

“For the quantification, the concentration of the MS-QBIC peptide (stable isotope-labeled peptide) was first determined by comparing the ion peak intensities of commercially available BSA.” The authors should state that they used BSA digested with trypsin.

The authors need to include the calibration curves they used to relate the signal intensities of the various peptides to one another. This should be included in the Supporting Information

“Nonamyloid and amyloid samples were differently labeled with light CH2O and heavy DH2O isotopes through dimethyl labeling with formaldehyde, respectively [24]” Do the authors mean CD2O or 13CD2O instead of DH2O?

Table 2: The lines separating proteins should be thicker than those used to delineate the peptides associated with that protein to avoid confusion. For example, the lines above and below B2M and its three peptides (VNHVTLSQPK, VEHSDLSFSK, and IQVYSR) should be thicker than the line used to separate VNHVTLSQPK and VEHSDLSFSK. In addition, there should be a line to separate VEHSDLSFSK, and IQVYSR.

Figure 4: The resolution is poor. The figure is unreadable.

References:

PLoS ONE requires full page numbers. The references need to be properly formatted.

Supporting Information Figure 1 and 2 are not available.

Reviewer #2: In this manuscript, authors applied mass spectrometry based quantification by isotope labeled cell free products (MS-QBIC) to formalin fixed, paraffin embedded (FFPE) tissues. The method was applied to amyloid tissues collected by laser capture microdissection of Congo red stained lesions of FFPE specimens. Authors managed to quantify 12 out of 13 selected proteins relevant to amyloidosis such as: serum amyloid A (SAA), transthyretin (TTR), immunoglobulin kappa light chain (IGK), immunoglobulin lambda light chain (IGL), beta-2 microglobulin (B2M), apolipoprotein (Apo) A1, Apo A2, Apo A4, Apo E, lysozyme, gelsolin, and fibrinogen alpha chain. Leukocyte cell derived chemotaxin-2 was not detected. Authors claim that quantification of amyloid protein by MS-QBIC is feasible and useful for the classification and research on systemic amyloidosis. The study although interesting requires further refinement to be considered for publication in Plos One.

Major Comments

1. The authors tried to quantify their MS-QBIC standard peptides with the help of the quantification tag from BSA which was included in all of the sequences of their MS-QBIC peptides. By trypsinizing a sample that included BSA of known concentration and the MS-QBIC peptide, the quantification tag of BSA was released (unlabeled-light from BSA and labeled-heavy from MS-QBIC peptide) and by comparing the signal intensity of light/heavy the concentration of the MS-QBIC peptide was determined. This is for sure an indirect method to quantify the standard peptides (MS-QBIC) that will be used later for the quantification of the selected target proteins. This method raised a lot of concerns since many data for this quantification are missing from the manuscript. Authors present only a supplementary figure (suppl. fig 2) which just describes the general workflow followed for the quantification of the MS-QBIC peptides but does not provide any data. Authors should present all the light (quantification tag from BSA) / heavy (quantification tag from MS-QBIC) ratios, the intensity values of each tag, the different known concentrations of BSA that were utilized in this analysis. Was it observed any saturation on the signal produced by the BSA tag and if yes at which concentration? It would be strongly recommended that the authors would test 6-8 different concentrations of BSA and select the values that are in the linear range. Light/Heavy ratios should be reported as well. Each sample of the BSA different concentration should be analyzed in triplicate and the authors should provide the mean value along with the standard deviation of the measurement. This is a very important step since based on these values the authors will better define the concentration-amount of their MS-QBIC isotope labeled peptides which will be used for the quantification of the selected target proteins.

2. Authors report the selected proteins, the precursors and the fragments measured for each protein with the SRM method. However, Light (endogenous) / Heavy (labeled standards) ratios are missing from the manuscript. Intensities of light, heavy and their respective ratios should be reported for all the quantified proteins along with the estimated concentration. Were the ratios obtained in the range of 0.1-10 (concentration balanced)? If not there is a big possibility that the reported concentrations are not accurate.

3. Authors report that more than three target peptides were quantified for each amyloid protein but it is not clear in the text if for the quantification the sum of the signals of all precursor ions was used or if the quantification was performed with the precursor ion that had the maximum intensity. Please elaborate on this. If the sum of all precursor ions intensity was used, then it would be highly recommended to provide the individual values of each precursor ion for each target protein as well. Authors should try to explain in more details the quantification workflow followed.

4. Standard curves for the peptides of interest are entirely missing from the manuscript. The authors should perform standard curves for the MS-QBIC peptides utilized in the present manuscript. Without the standard curve how did they estimate the appropriate amount of MS-QBIC peptide that had to be spiked in to the sample to provide a concentration balanced ratio (Light/Heavy ratio in the range of 0.1-10). Without this information it is really difficult to estimate the appropriate amount of the heavy standard to be spiked into the sample. If the Light/Heavy ratio is not in the range of 0.1-10 then the reported values are not reliable. With the standard curve, the ideal concentration of the heavy labeled peptide to be added into the sample is estimated in order to have a more accurate result. This ideal concentration is the "concentration balanced" value where the Light/Heavy ratio is in the range of 0.1-10. In the present study authors used 3 peptides per target protein. Ideally, standard curves of all the peptides should be prepared. If not possible for all the peptides per target protein then at least for the one with the maximum signal intensity for each protein. 6-8 different concentrations of the MS-QBIC peptide should be used for the standard curve.

5. Along the same lines, authors should try to define the Lower Limit Of Detection (LLOD) and the Lower Limit Of Quantification (LLOQ) for the proteins reported in the manuscript.

6. Concentration values reported in pmol/mm2 is not very helpful. It would be better to define the pmol for each target protein per sample. The FFPE area that was utilized for the targeted proteomics analysis was identical for each sample? Are there any reference values from the literature with regards to the selected target proteins quantified? If yes, please compare the reference values with the ones obtained from this study.

7. Please provide some representative figures showing the precursor and the fragments (transitions) spectra for each one of the target proteins for all the samples analyzed. This can be submitted as supplementary information. Peak-shape, retention time, quality of the spectra and co-elution of endogenous and isotope labeled peptide transitions should be clearly shown in this figures.

8. Please provide any relative information regarding the purity of the MS-QBIC peptides utilized in the study. Was the purity somehow estimated and if yes what method was applied?

9. It would be helpful for the readers and very beneficial for the scientific community if the raw files were submitted as supplementary material or if they were deposited in an open access public repository such as PRIDE or Proteome Xchange.

Minor Comments

1. It is not very clear if the MS-QBIC (isotope labeled peptides) were spiked into the sample from the begining of the sample preparation process or if they were added after the trytpic digestion of the samples. Please specify.

2. In the section: "Target peptides of amyloid proteins" (page 8 last paragraph) please provide some more information with regards to the PD settings utilized such as: mass tolerance (for the precursor and fragment), modifications (static and/or dynamic), FDR. It is also not clear in the text if the authors performed reduction and alkylation during the sample preparation. Please elaborate on this.

3. What was the trypsin/protein ratio utilized for the tryptic digestion? How many ng of trypsin were utilized per sample?

6. PLOS authors have the option to publish the peer review history of their article (what does this mean?). If published, this will include your full peer review and any attached files.

Reviewer #1: No

Reviewer #2: No

---

## [Author Response · Author response to Decision Letter 0]

13 May 2020

PONE-D-20-02311

Mass spectrometry–based absolute quantification of amyloid proteins in pathology tissue specimens: merits and limitations

Responses to Reviewers’ Comments

We would like to thank both reviewer for their thorough reading of our manucript. We greatly appreciate the insightful comments and useful suggestions.

Before our point-to-point responses, we would like to list the major revisions that were made in accordance with the reviewers’ suggestions.

A: Methods (analysis of MS)

1) Calibration curves of BSA: We added the calibration curves of BSA in a supporting figure (S3 A Fig.). The calibration curve shows that the signal increases almost linearly with the increase in concentration of BSA.

2) Heavy/light ratios of MS-QBIC peptides: The concentrations of the synthesized MS-QBIC peptides were determined by analyzing 100 fmol of digested BSA (light-labeled: known concentration) and co-injected BSA-tag of the MS-QBIC peptides (heavy-labeled). For the initial quantification of the synthesized MS-QBIC products, all the heavy/light ratios of the BSA-quantification tags were within the range of the calibration curve and within the range of 0.1–10 heavy (MS-QBIC peptide)/light (endogenous amyloid peptide) ratios (S3 B Fig.).

4) Signal linearity of each MS-QBIC product: The signal lineality of each MS-QBIC product was evaluated by analyzing different amounts of MS-QBIC peptides (S2 Appendix).

5) Lower Limit of Quantification (LLOQ): We have defined the LLOQ amount of each MS-QBIC peptide by determining the value of the minimum amount tested for which the intensity was within a 30% difference of the linear regression line obtained for each calibration.

6) Criteria for selecting MS-QBIC peptides for the absolute quantification: MS-QBIC peptides were excluded from the analysis when (1) the MS-QBIC peptides were not able to be synthesized, (2) the MS-QBIC peptides failed to have a linear regression line that could be fitted to ≥4 consecutive amounts with R2>0.95 and a % difference for each quantification point <30, (3) the MS-QBIC peptides could not be detected robustly after mixing with laser microdissection (LMD) samples, and (4) the peptides could not be separated from other adjacent signals in the LMD sample, or could not be detected in the samples.

B: Data presentation

1) Raw data of amyloid peptides: We have included endogenous/labeled-standard ratios for the quantification of all the peptides in a supplementary table (S3 Table). The quantitative value for each amyloid precursor protein is presented as the quantified value (pmol) of the protein divided by the volume (mm3) that was resected from each sample via LMD. We used the average of the quantitative values of three samples of the amyloid protein from the same sections for further analyses.

2) Raw data of SRM analysis: The raw data of SRM analysis were deposited to the PeptideAtlas SRM Experiment Library (PASSEL) (http://www.peptideatlas.org/passel/). (Note that we have set the password “msqbic” during the peer-review process of this manuscript.)

3) Chromatogram/spectrum: The precursor and fragment ions chromatogram/spectrum for a representative peptide of all the quantified amyloid proteins (excluding LECT2 that we could not detect in this study) are presented in S3 Appendix.

C. Results

According to the refinements in the absolute quantification described above, the data of the MS analysis were re-evaluated.

1) Approximately 80% of the quantified amount was calculated within the range of 0.1–10 heavy (MS-QBIC peptide)/light (endogenous amyloid peptide) ratios, indicating that in most cases, the concentrations of the spiked peptide and the quantification targets were well balanced. The remaining ~20% of quantification values were calculated within the range of 0.01–100 heavy/light ratios.

2) A total of 4971 quantified values were used for the absolute quantification, instead of 8177 in the original analysis. Because most of the excluded values were trace amounts, this filtration of values did not affect the original results, except for lysozyme. We excluded ‘WESGYNTR’ from the peptides of lyzsozyme because the re-evaluation indicated that the peak for this peptide was not clearly separated. This exclusion considerably lowered the estimated amounts of lysozyme, most of which were below the minimum amounts of the major amyloid proteins.

D. Figure and Tables

Tables 2–6 were revised according to the re-evaluation of data.

We added the following supporting information.

S2 Appendix. Signal linearity of each MS-QBIC peptide measured by mass spectrometry

S3 Appendix. Precursor and fragment ions chromatogram/spectrum for a representative peptide of the quantified amyloid proteins

S3 Fig. Supporting information for the estimation of the concentration of MS-QBIC peptides

S2 Table. List of MS-QBIC target peptides, primer sequences for the production of MS-QBIC peptides, and SRM method for the quantification.

S3 Table. List of information used for the absolute quantification of amyloid proteins in the tissue samples, including the intensities and heavy/light ratios of MS-QBIC- and tissue-derived peptide signals.

The reviewers’ comments are italicized and enclosed in square brackets. Modifications to the revised manuscript are highlighted in red.

[Reviewer #1]

[The authors have applied their previously published MS-QBIC to quantitate the amyloids present in FFPE tissue samples. Their skill using mass spectrometry, pathology, and microdissection is obvious. They used PTS method to extract the proteins from the FFPE tissues. PTS is most appropriate for unfixed tissue.

Formalin fixing uses formaldehyde to crosslink proteins by their lysine and other residues. The authors used stable isotope-labeled formaldehyde in this work and they ensured that the consequent imine formation was reduced using sodium cyanoborohydride. The crosslinking resulting from formalin fixing would make some proteins inaccessible to trypsin and prevent some trypsin cleavage on proteins that were accessible to trypsin. The authors used the PTS method to extract proteins from this fixed tissue. This would assist in extracting the proteins, but it does not undo the formaldehyde mediated cross linking of the formalin. Other researchers use high temperatures and free amines to reverse the fixation process and yield intact proteins.]

As the reviewer pointed out, we actually heated FFPE samples to reverse the formaldehyde cross-links. The revised method section describes the details of the sample preparation:

(line 173)

All the samples were digested to recover the peptides according to the phase-transfer surfactant (PTS) method [8, 19] with several modifications. In brief, samples were dissolved in PTS buffer [8, 19] and heated at 95°C for 3 h to reverse formalin cross-linking.

(line 314)

Samples collected in PTS buffer were heated at 95°C for 3 h to reverse formalin cross-linking prior to the reducing step with 100 mM dithiothreitol.

(line 374)

Samples collected in PTS buffer were heated at 95°C for 3 h to reverse formalin cross-links prior to the reducing step with 100 mM dithiothreitol.

[The authors need to revise the manuscript by including an additional limitation: namely that fixation may not be uniform for each protein. This may be why the authors were able to observe some tryptic peptides and not others. It also means that discussions about quantitation of proteins makes no sense when the authors cannot ensure that the proteins they are quantitating are extracted in proportion to their prevalence.]

We appreciate the insightful comments of the reviewer. We commented on this important issue in the Discussion.

(line 678-688)

Absolute quantification is potentially useful to compare the various effects of different types of amyloid in the diseased tissue. However, there are limitations in the present study. The effect of formalin fixation might not be uniform at each amino acid residue in a single protein, which may affect the accuracy of quantification afforded by a limited number of peptide candidates for one amyloid protein. Second, the amount of amyloid deposits was evaluated in units of pmol/mm3 with the volume of amyloid as the denominator. This evaluation ignores the spatial distribution of amyloid proteins within the deposit area. However, the physical properties of each amyloid protein might be different if, for example, high concentrations of a particular type of amyloid protein are deposited in a particular region of the deposit area.

Some specific changes are listed below.

[The authors use acronyms such as PTS (page 8) before they define it as phase transfer surfactants (page 10). They should carefully check the manuscript to ensure that the acronyms are defined before they are used.]

We have carefully checked the manuscript to ensure that the acronyms are all defined at first mention. We revised our manuscript as follows:

(line172)

All the samples were digested to recover the peptides according to the phase-transfer surfactant (PTS) method.

[The statement: “The various forms of amyloidosis require different approaches, including high-dose…” Should be revised to include “treatment”: “The various forms of amyloidosis require different treatment approaches, including high-dose…”]

We revised our manuscript as follows: 

(line 58)

The various forms of amyloidoses require different treatment approaches, such as high-dose melphalan with stem cell transplantation in AL amyloidosis and liver transplantation or use of transthyretin (TTR)-tetramer stabilizer in hereditary TTR amyloidosis.

[“13C6 15N4 L-arginine and 13C6 L-lysine.” Is Lysine unlabeled with 15N?]

Yes. Lysine is not labeled with 15N. (line 230)

[“For the quantification, the concentration of the MS-QBIC peptide (stable isotope-labeled peptide) was first determined by comparing the ion peak intensities of commercially available BSA.” The authors should state that they used BSA digested with trypsin.]

The undigested MS-QBIC peptide and undigested BSA were mixed and then digested with trypsin. We revised our manuscript as follows:

(line 242)

MS-QBIC product was then mixed with BSA and the mixture was subjected to trypsin digestion by following to the PTS method as described above.

[The authors need to include the calibration curves they used to relate the signal intensities of the various peptides to one another. This should be included in the Supporting Information]

We have included S3 Fig. (the calibration curves of BSA, and the MS-QBIC peptides and samples). The linearity of the MS-QBIC peptides is presented in S2 Appendix.

[“Nonamyloid and amyloid samples were differently labeled with light CH2O and heavy DH2O isotopes through dimethyl labeling with formaldehyde, respectively [24]” Do the authors mean CD2O or 13CD2O instead of DH2O?]

As the reviewer has pointed out, CH2O was incorrectly described. We revised our manuscript as follows:

(line 363)

Nonamyloid and amyloid samples were differently labeled with light CD2O and heavy 13CD2O isotopes through dimethyl labeling with formaldehyde, respectively.

[Table 2: The lines separating proteins should be thicker than those used to delineate the peptides associated with that protein to avoid confusion. For example, the lines above and below B2M and its three peptides (VNHVTLSQPK, VEHSDLSFSK, and IQVYSR) should be thicker than the line used to separate VNHVTLSQPK and VEHSDLSFSK. In addition, there should be a line to separate VEHSDLSFSK, and IQVYSR.]

We revised Table 2 in accordance with the reviewer’s comments.

[Figure 4: The resolution is poor. The figure is unreadable.]

We have revised Figure 4 with higher resolution.

[PLoS ONE requires full page numbers. The references need to be properly formatted.]

We have added “full page numbers” for the references.

[Supporting Information Figure 1 and 2 are not available.]

Supporting information has been made available according to the PLoS One rules.

[Reviewer #2]

[Major Comments]

[1. The authors tried to quantify their MS-QBIC standard peptides with the help of the quantification tag from BSA which was included in all of the sequences of their MS-QBIC peptides. By trypsinizing a sample that included BSA of known concentration and the MS-QBIC peptide, the quantification tag of BSA was released (unlabeled-light from BSA and labeled-heavy from MS-QBIC peptide) and by comparing the signal intensity of light/heavy the concentration of the MS-QBIC peptide was determined. This is for sure an indirect method to quantify the standard peptides (MS-QBIC) that will be used later for the quantification of the selected target proteins. This method raised a lot of concerns since many data for this quantification are missing from the manuscript. Authors present only a supplementary figure (suppl. fig 2) which just describes the general workflow followed for the quantification of the MS-QBIC peptides but does not provide any data.　 Authors should present all the light (quantification tag from BSA) / heavy (quantification tag from MS-QBIC) ratios, the intensity values of each tag, the different known concentrations of BSA that were utilized in this analysis. Was it observed any saturation on the signal produced by the BSA tag and if yes at which concentration? It would be strongly recommended that the authors would test 6-8 different concentrations of BSA and select the values that are in the linear range. Light/Heavy ratios should be reported as well. Each sample of the BSA different concentration should be analyzed in triplicate and the authors should provide the mean value along with the standard deviation of the measurement. This is a very important step since based on these values the authors will better define the concentration-amount of their MS-QBIC isotope labeled peptides which will be used for the quantification of the selected target proteins.]

We agree with the reviewer’s concerns. We have added the calibration curves of BSA in S3 Fig. The calibration curve shows that the signal increases almost linearly with increasing concentrations of BSA. The concentrations of the synthesized MS-QBIC peptides were determined by analyzing 100 fmol of digested BSA (light-labeled: known concentration) and co-injected quantification-tags of the MS-QBIC peptides (heavy-labeled). An advantage of stable isotope-based quantification is that this method works even in the presence of appreciable levels of ion suppression (i.e., when the calibration curve is no longer linear). Nonetheless, for the initial quantification of these synthesized MS-QBIC products, all the heavy/light ratios of the BSA-quantification tags were withing the range of the calibration curve and within the range of 0.1–10 heavy (MS-QBIC peptide)/light (endogenous amyloid peptide) ratios. Therefore, we estimated the amount of MS-QBIC peptide to be proportional to the quantified heavy/light ratio. We determined the MS-QBIC peptide concentration based on a single analysis; as the reviewer has pointed out, this may limit the accuracy of our quantification values for each MS-QBIC peptide. However, we used different peptides for the quantification of a single amyloid protein, and the quantified values were overall consistent between the different peptides used to quantify the same amyloid protein. Thus, we believe that our quantification well represents the amount of deposited amyloid protein in each microdissected tissue.

To explain these processes, we have revised our manuscript as follows:

(line 232-282)

The concentration of the MS-QBIC peptide (stable isotope-labeled peptide) was first determined by comparison with the ion peak intensity of commercially available bovine serum albumin (BSA) as follows. First, weighed BSA powder (≥98.0% purity, Sigma-Aldrich) was dissolved in PTS buffer. The MS-QBIC product was then mixed with BSA and the mixture was subjected to trypsin digestion according to the PTS method as described above except 100 mM tris (2-carboxyethyl) phosphine hydrochloride (Nacalai Tesque, Kyoto, Japan) was used instead of 100 mM dithiothreitol. The pre-heating procedure was not applied to digest the non-fixed BSA/MS-QBIC samples. To reduce the non-specific binding of peptides to sample tubes, all the processes were carried out in the presence of 1 pmol/μL of trypsin-digested α-enolase (Sigma-Aldrich) prepared according to the PTS method. The signal linearity of the quantification tag (a peptide sequence derived from BSA: S2 Figure) was confirmed by analyzing different amounts of BSA (S3A Figure), although the stable isotope-based quantification does not heavily depend on the linearity of the signal with an increase in the peptide amount [25].

For the quantification of the tag sequence and all MS-QBIC-based quantification, it was carried out by selected reaction monitoring (SRM) analysis using a TSQ Quantiva triple-stage quadrupole mass spectrometer (Thermo Fisher Scientific). The following parameters were selected: positive mode, scan width of 0.002 m/z, Q1 and Q3 resolutions of 0.7 full width of half maximum (FWHM), cycle time of 3 s, and gas pressure of 1.8 Torr. The spectrometer was equipped with an UltiMate 3000 RSLCnano nano-high performance liquid chromatography (HPLC) system (Thermo Fisher Scientific), and a PepMap HPLC trap column (C18, 5 µm, 100 Å; Thermo Fisher Scientific) for loading samples. Analytical samples were solubilized in 2% acetonitrile and 0.1% TFA, and separated by reversed-phase chromatography using a PepMap rapid separation liquid chromatography (RSLC) EASY-Spray column (C18, 3 µm, 100 Å, 75 µm x 15 cm; Thermo Fisher Scientific) using mobile phases A (0.1% formic acid/H2O) and B (0.1% formic acid and 100% acetonitrile) at a flow rate of 300 nL/min (4% B for 5 min, 4% to 36% B in 20 min, 36% to 95% B in 1 min, 95% B for 5 min, 95% to 4% B in 1 min and 4% B for 13 min). The eluted material was directly electro-sprayed into the MS. SRM transitions of the target peptides were optimized using Pinpoint software, version 1.3 (Thermo Fisher Scientific) (S2 Table). The Quan Browser of the Xcalibur data system, version 2.2 (Thermo Fisher Scientific) was used for data processing. The calibration curve of the BSA-derived quantification tag (S3 A Fig) shows linear increase of signal intensity according to the increase of injected BSA; thus the amount of each MS-QBIC peptide was estimated to be proportional to the quantified heavy (MS-QBIC product) to light (weighed BSA) ratio. To determine the stock concentration of the MS-QBIC peptide, 0.2 µL of stock solution containing 100 fmol of digested BSA was analyzed. All the quantifications were within the range of 0.1-10 heavy (quantification tag derived from MS-QBIC peptide) to light (the tag derived from digested BSA) ratios (S3 B Fig).

[2. Authors report the selected proteins, the precursors and the fragments measured for each protein with the SRM method. However, Light (endogenous) / Heavy (labeled standards) ratios are missing from the manuscript. Intensities of light, heavy and their respective ratios should be reported for all the quantified proteins along with the estimated concentration. Were the ratios obtained in the range of 0.1-10 (concentration balanced)? If not there is a big possibility that the reported concentrations are not accurate.] 

We have included the endogenous to labeled standard ratios for all quantifications in a supplementary table (S3 Table). As indicated, many quantifications were carried out within the range of 0.1-10. However, it is technically difficult to control the amount of spiked peptide over 10,000 quantifications, and approximately 20% of the quantification processes were carried out outside of the range of 0.1-10 (but still within a 0.01-100 ratio). In the revised manuscript, we have also defined the LLOQ (please see our response below) and excluded quantified values below the LLOQ. Therefore, again, we believe that our quantification well represents the amount of deposited amyloid protein in each microdissected tissue.

The corresponding parts of the revised manuscript are as follows:

(lines 326-343)

A TSQ Quantiva triple-stage quadrupole mass spectrometer (Thermo Fisher Scientific) was used for SRM analysis as described above. The target peptide derived from the amyloid sample was quantified by comparing the intensity of the nonlabeled target peptide in the sample with that of the MS-QBIC peptide of known concentration, which had been added to the sample prior to the measurement.

At least two or more target peptides were quantified per amyloid precursor protein. The quantitative value of each amyloid precursor protein was presented as the quantified value (pmol) of the protein divided by the volume (mm3) that was resected from each sample via LMD. We used the average of the quantitative value of the amyloid protein for three samples of the same specimen for further analyses. S3 Table summarizes the raw quantification values. Apporximately 80% of the quantified amount was calculated within the range of 0.1-1 0 heavy (MS-QBIC peptide) to light (endogenous amyloid peptide) ratios, indicating that in most cases, the concentrations of the spiked peptide and the quantification targets were well balanced. The remaining ~20% of quantification values were calculated within the range of 0.01-100 heavy to light ratios.

[3. Authors report that more than three target peptides were quantified for each amyloid protein but it is not clear in the text if for the quantification the sum of the signals of all precursor ions was used or if the quantification was performed with the precursor ion that had the maximum intensity. Please elaborate on this. If the sum of all precursor ions intensity was used, then it would be highly recommended to provide the individual values of each precursor ion for each target protein as well. Authors should try to explain in more details the quantification workflow followed.]

We appreciate the reviewer’s comment. In the present study, we used the sum of the signals (average) instead of the maximum intensity. Further evaluation will be necessary in the future to determine the best method for the quantification of different amyloid proteins. We have presented the raw data in S3 Table.

[4. Standard curves for the peptides of interest are entirely missing from the manuscript. The authors should perform standard curves for the MS-QBIC peptides utilized in the present manuscript. Without the standard curve how did they estimate the appropriate amount of MS-QBIC peptide that had to be spiked into the sample to provide a concentration balanced ratio (Light/Heavy ratio in the range of 0.1-10). Without this information it is really difficult to estimate the appropriate amount of the heavy standard to be spiked into the sample. If the Light/Heavy ratio is not in the range of 0.1-10 then the reported values are not reliable. With the standard curve, the ideal concentration of the heavy labeled peptide to be added into the sample is estimated in order to have a more accurate result. This ideal concentration is the "concentration balanced" value where the Light/Heavy ratio is in the range of 0.1-10. In the present study authors used 3 peptides per target protein. Ideally, standard curves of all the peptides should be prepared. If not possible for all the peptides per target protein then at least for the one with the maximum signal intensity for each protein. 6-8 different concentrations of the MS-QBIC peptide should be used for the standard curve.]

[5. Along the same lines, authors should try to define the Lower Limit Of Detection (LLOD) and the Lower Limit Of Quantification (LLOQ) for the proteins reported in the manuscript.]

We have included a description of our evaluation method of the accuracy of our quantifications that we believe addresses the concerns #4 and #5. We evaluated the linearity of the intensity of different concentrations of MS-QBIC peptides for all the sequences. In the revised manuscript, we have used only the MS-QBIC sequences with acceptable calibration curves, that is only those that meet the revised criteria. We have further defined the LLOQ amount of each MS-QBIC peptide by determining the value of the minimum tested amount for which the intensity was within a 30% difference of the linear regression line obtained for each calibration.

By incorporating these evaluations, we analyzed the data again, and obtained results as follows:

1) Approximately 80% of the quantified amount was calculated within the range of 0.1-10 heavy (MS-QBIC peptide) to light (endogenous amyloid peptide) ratios, indicating that in most cases, the concentrations of the spiked peptide and the quantification targets were well balanced. The remaining ~20% of quantification values were calculated within the range of 0.01-100 heavy to light ratios.

2) This filtration of values did not affect the original results, except for lysozyme. We excluded ‘WESGYNTR’ from the peptides of lyzsozyme because the re-evaluation indicated that the peak for this peptide was not clearly separated. This exclusion considerably lowered the estimated amounts of lysozyme, most of which were below the minimum amounts of the major amyloid proteins.

The corresponding parts of revised manuscript are as follows:

(lines 283-301)

Confirmation of signal linearity for each MS-QBIC product: The signal linearity for each MS-QBIC product was then evaluated by analyzing different amounts of MS-QBIC peptides (S2 Appendix). For the linear regression evaluation, we excluded MS-QBIC peptides that failed to have the linear regression line that could be fitted to ≥4 consecutive amounts with R2>0.95 and a % difference for each amount <30. The MS-QBIC peptides were further investigated to determine if the peptides could be detected robustly in the presence of peptides recovered from tissue samples. Several test tissue samples were prepared and mixed with MS-QBIC peptides according to the method described in the next section. A few MS-QBIC peptides that could not be detected robustly in the presence of tissue samples were excluded for the absolute quantification steps.

Lower Limit of Quantification: Next, based on the liner regression analysis, we defined the Lower Limit of Quantification (LLOQ) by determining the minimum amount of each MS-QBIC peptide for which the intensity was within a 30% difference of the linear regression line obtained in the each calibration plot. If all analyzed amounts were within the 30% difference, the minimum amount used to produce the linear regression line was set as the LLOQ. The values of the LLOQ are shown in S3 Table.

[6. Concentration values reported in pmol/mm2 is not very helpful. It would be better to define the pmol for each target protein per sample. The FFPE area that was utilized for the targeted proteomics analysis was identical for each sample? Are there any reference values from the literature with regards to the selected target proteins quantified? If yes, please compare the reference values with the ones obtained from this study.]

In the revised manuscript, we present the estimated concentration of each peptide in units of pmol/mm3. The supplementary table S3 includes the volume of the dissected samples (in mm3) so that readers can convert the concentrations to pmol depending on their interest.

We commented on this topic (limitations) in the Discussion.

(lines 683-688)

Second, the amount of amyloid deposits was evaluated in units of pmol/mm3 with the volume of amyloid as the denominator. This evaluation ignores the spatial distribution of amyloid proteins within the deposit area. However, the physical properties of each amyloid protein might be different if, for example, high concentrations of a particular type of amyloid protein are deposited in a particular region of the deposit area.

[7. Please provide some representative figures showing the precursor and the fragments (transitions) spectra for each one of the target proteins for all the samples analyzed. This can be submitted as supplementary information. Peak-shape, retention time, quality of the spectra and co-elution of endogenous and isotope labeled peptide transitions should be clearly shown in this figures.]

[8. Please provide any relative information regarding the purity of the MS-QBIC peptides utilized in the study. Was the purity somehow estimated and if yes what method was applied?]

In accordance with the reviewer’s advice, points #7 and #8, we have presented the precursor and fragment ions chromatograms/spectra for the representative peptides.

The peak shape provided the relevant information regarding the purity of the MS-QBIC peptides.

The revised manuscript includes this information as follows:

(lines 344-354)

The chromatogram of precursor ion and the spectrum of fragments of ions for a representative peptide for all the quantified amyloid proteins (excluding LECT2 that we could not detect in this study) are presented in S3 Appendix. The chromatograms of the endogenous and isotope-labeled peptides indicated that the two co-eluted peaks (endogenous and MS-QBIC peptides) were clearly detected and almost no overlap with other peaks from the background occurred. This peak shape provided information regarding the purity of the MS-QBIC peptides, indicating that possible contaminants or unexpected sub-products of the MS-QBIC peptides made a negligible contribution to the quantification of selective ion monitoring using the triple quadrupole mass spectrometer.

[9. It would be helpful for the readers and very beneficial for the scientific community if the raw files were submitted as supplementary material or if they were deposited in an open access public repository such as PRIDE or Proteome Xchange.]

PRIDE may not be the best place to deposit a SRM/MRM dataset without information regarding MS/MS-based protein identification. We have uploaded the raw data to the PeptideAtlas.

(line 354-357)

The raw data from the SRM analysis were deposited to the Peptide Atlas SRM Experiment Library (PASSEL; http://www.peptideatlas.org/passel/; dataset identifier: PASS01558; see S3 Table for the relationship between raw file names and quantified results).

Note that we have set the password “msqbic” during the peer-review process of this manuscript.

[Minor Comments]

[1. It is not very clear if the MS-QBIC (isotope labeled peptides) were spiked into the sample from the begining of the sample preparation process or if they were added after the trytpic digestion of the samples. Please specify.]

The MS-QBIC peptides digested by trypsin were spiked into the amyloid samples after the tryptic digestion of samples.

We have specified this point as follows:

(lines 320-323)

The digested samples were desalted using a self-prepared C18 stage tip and dissolved in water containing 2% acetonitrile and 0.1% TFA. The MS-QBIC peptide mixture was added to the sample at this stage and then the sample was subjected to the MS analysis.

[2. In the section: "Target peptides of amyloid proteins" (page 8 last paragraph) please provide some more information with regards to the PD settings utilized such as: mass tolerance (for the precursor and fragment), modifications (static and/or dynamic), FDR. It is also not clear in the text if the authors performed reduction and alkylation during the sample preparation. Please elaborate on this.]

We have revised our Methods section to fully describe the settings of Proteome Discoverer as follows:

(lines 173-204):

All the samples were digested to recover the peptides according to the phase-transfer surfactant (PTS) method [8, 19] with several modifications. In brief, the sample was dissolved in PTS buffer [8, 19] and heated at 95 °C for 3 h to reverse formalin cross-links. The sample was reduced with 100 mM dithiothreitol (FUJIFILM Wako Pure Chemical Corp., Tokyo, Japan) at room temperature for 30 min, and then alkylated with 1 M iodoacetamide (Sigma-Aldrich) at room temperature for 30 min. Next, the sample was digested by adding 1 μg of lysyl endopeptidase (Lys-C) (FUJIFILM Wako Pure Chemical Corp.). After incubating the samples at 37 °C for 3 h, 1 μg trypsin (Roche) was added and the mixture was further incubated at 37 °C overnight. The detergents were removed by ethyl acetate/trifluoroacetic acid (TFA) solution according to the PTS method. The digested samples were then desalted using a self-prepared C18 stage tip [20]. The peptides were solubilized in 2% acetonitrile and 0.1% TFA and loaded to the LC-MS system to be separated by a gradient using mobile phases A (0.1% formic acid/H2O) and B (0.1% formic acid and 100% acetonitrile) at a flow rate 300 nL/min (4% to 36% B in 20 min, 36% to 95% B in 1 min, 95% B for 5 min, 95% to 4% B in 1 min, and 4% B for 18 min) with a home-made capillary column (length of 200 mm and inner diameter of 100 μm) packed with 3μm C18 resin (L-column2, Chemicals Evaluation and Research Institute, Japan). The eluted peptides were electrosprayed (1.8-2.3 kV) and introduced into the Q-Exactive MS equipment (Thermo Fisher Scientific) in positive ion mode with data-dependent MS/MS. The obtained raw data was subjected to database search (UniProt, reviewed human database as of May 7th 2014) with the Sequest HT algorithm running on Proteome Discoverer 1.4 (Thermo Fisher Scientific). The parameters for database searches were as follows. Peptide cleavage was set to trypsin. Missed cleavage was allowed up to 2 and minimum and maximum peptide length was 6-144 amino acids. The mass tolerances were set to 10 ppm for precursor ions and 0.02 Da for fragment ions. For modification conditions, carbamidomethylation at cysteine was set as fixed modification and oxidation at methionine was set as variable modification. A significance threshold of P<0.05 was applied.

[3. What was the trypsin/protein ratio utilized for the tryptic digestion? How many ng of trypsin were utilized per sample?]

We collected 0.0007-0.0027 mm3 volumes of amyloid deposits. If the amount of protein is approximately 10% of the wet weight of the tissue sample, then the 0.0007–0.0027 mm3 samples should contain 0.7–2.7 µg of protein. We added 1 µg of trypsin into each sample, thus the amount of trypsin was assumed to be sufficient to digest all the proteins.

The revised Methods section now reads:

(lines 312-323)

Enzymatic digestion of the 174 amyloid samples and MS-QBIC peptides was performed essentially according to the PTS protocol as previously described above. Samples collected in PTS buffer were heated at 95°C for 3 h to reverse the formalin cross-links, prior to the reducing step with 100 mM dithiothreitol. A 0.0007-0.0027 mm3 volume of collected amyloid sample may contain 0.7-2.7 mg of protein, assuming that the amount of protein is approximately 10% of the wet weight of the tissue sample. Thus, the addition of 1 µg of Lys-C/trypsin into each sample should be sufficient to digest all the amyloid proteins. The digested samples were desalted using a self-prepared C18 stage tip and dissolved in water containing 2% acetonitrile and 0.1% TFA. The MS-QBIC peptide mixture was added to the sample at this stage and then the sample was subjected to the MS analysis.

Again, thank you for giving us the opportunity to strengthen our manuscript with your valuable comments and queries. We have worked hard to incorporate your feedback and hope these revisions will make this manuscript acceptable for publication.

---

## [Decision Letter · Decision Letter 1]

3 Jun 2020

PONE-D-20-02311R1

Mass spectrometry–based absolute quantification of amyloid proteins in pathology tissue specimens: merits and limitations

PLOS ONE

Dear Dr. Fukayama,

Thank you for submitting your manuscript to PLOS ONE. After careful consideration, we feel that it has merit but does not fully meet PLOS ONE’s publication criteria as it currently stands. Therefore, we invite you to submit a revised version of the manuscript that addresses the points raised during the review process.

We look forward to receiving your revised manuscript.

Kind regards,

Harald Mischak

Academic Editor

PLOS ONE

Additional Editor Comments (if provided):

The paper is in principle accceptable, but a few corrections are required, as indicated by reviewer 1. Please implement these minor changes and resubmit, so the paper can be accepted

Reviewers' comments:

Reviewer's Responses to Questions

**Comments to the Author**

1. If the authors have adequately addressed your comments raised in a previous round of review and you feel that this manuscript is now acceptable for publication, you may indicate that here to bypass the “Comments to the Author” section, enter your conflict of interest statement in the “Confidential to Editor” section, and submit your "Accept" recommendation.

Reviewer #1: (No Response)

Reviewer #2: All comments have been addressed

2. Is the manuscript technically sound, and do the data support the conclusions?

Reviewer #1: Yes

Reviewer #2: Yes

3. Has the statistical analysis been performed appropriately and rigorously? 

Reviewer #1: Yes

Reviewer #2: Yes

4. Have the authors made all data underlying the findings in their manuscript fully available?

Reviewer #1: Yes

Reviewer #2: Yes

5. Is the manuscript presented in an intelligible fashion and written in standard English?

Reviewer #1: Yes

Reviewer #2: Yes

6. Review Comments to the Author

Reviewer #1: Dear Editor and Authors,

The authors have revised the manuscript in accordance with this reviewer’s requested. In addition, they have also made other corrections not noted by the reviewer. There are some minor corrections that should be made. This reviewer does not need to see them.

Line 203, 429: The authors refer to the Q-Exactive mass spectrometer as “equipment”. The term instrument is more commonly used.

Line 410: “CD2O and heavy 13CD2O isotopes, respectively, through dimethyl labeling with” Should it be “CH2O and heavy 13CD2O isotopes, respectively, through dimethyl labeling with” Are the authors using CH2O (+0 Da), CD2O (+2 Da), and 13CD2O (+3 Da) or CH2O (+0 Da) and 13CD2O (+ 3 Da) in their analysis? Lines 424-425 suggest the latter is the Authors’ intent.

Line 412: “The interstitial tissues of 11 specimens without amyloidosis” “interstitial” seem to be the incorrect word for the sentence.

Lines 450-456: 6+7+2+5+6 =26 Should it be 26 of 30 or 24 of 30, but some patients had two forms? Compare with lines 502-507: 7+7+4+5+1=24.

Lines 606-608: An awkward sentence that should be revised: “For setting the the cut-off value for MS-quantification in the present study, we used a simulation to obtain the cut-off value as shown in Table 4.”

Lines 414, 592, 724: It is Sjögren or Sjögren’s syndrome

Reviewer #2: The authors have adequately addressed the comments raised. Revised manuscript can be now considered for publication.

7. PLOS authors have the option to publish the peer review history of their article (what does this mean?). If published, this will include your full peer review and any attached files.

Reviewer #1: No

Reviewer #2: No

---

## [Author Response · Author response to Decision Letter 1]

8 Jun 2020

Responses to Reviewers’ Comments

We greatly appreciate the critical comments and useful suggestions for our submission from reviewer #1.

The reviewer’s comments are italicized and enclosed in square brackets. Modifications in the revised manuscript are highlighted in red.

Line 203, 429: The authors refer to the Q-Exactive mass spectrometer as “equipment”. The term instrument is more commonly used.]

As the reviewer pointed out, we have corrected “equipment” to “instrument” in

Revised manuscript, Line 197, and Revised manuscript, Line 386.

[Line 410: “CD2O and heavy 13CD2O isotopes, respectively, through dimethyl labeling with” Should it be “CH2O and heavy 13CD2O isotopes, respectively, through dimethyl labeling with” Are the authors using CH2O (+0 Da), CD2O (+2 Da), and 13CD2O (+3 Da) or CH2O (+0 Da) and 13CD2O (+ 3 Da) in their analysis? Lines 424-425 suggest the latter is the Authors’ intent.

We are very sorry for our mistake, and we have corrected “CD2O” to “CH2O”:

Revised manuscript, Line 367

Nonamyloid and amyloid samples were differently labeled with light CH2O and heavy CD2O isotopes, respectively

[Line 412: “The interstitial tissues of 11 specimens without amyloidosis” “interstitial” seem to be the incorrect word for the sentence.]

As the reviewer pointed out, we changed it as follows; 

Revised manuscript, Line 369

The stromal components of 11 specimens devoid of amyloid (derived from seven patients with conditions unrelated to amyloidosis) were micro-dissected using LMD.

[Lines 450-456: 6+7+2+5+6 =26 Should it be 26 of 30 or 24 of 30, but some patients had two forms? Compare with lines 502-507: 7+7+4+5+1=24.]

We are very sorry for the confusing description.

Revised manuscript, Lines 406-412: 

In 24 of the 30 patients, amyloidosis was subclassified by clinical and pathological examination at autopsy, based on immunohistochemical criteria. Seven patients were AA amyloidosis. Of the seven patients with ATTR/TTR amyloidosis, two patients had hereditary disease, and five patients had acquired disease. Six patients had associated multiple myeloma, and three patients were diagnosed as primary or systemic amyloidosis. One patient had dialysis related amyloidosis.

[Lines 606-608: An awkward sentence that should be revised: “For setting the cut-off value for MS-quantification in the present study, we used a simulation to obtain the cut-off value as shown in Table 4.”]

We are very sorry for the awkward sentence. We change the sentence as follows.

Revised manuscript, Lines 408-415: 

For setting the cut-off value for MS-quantification in the present study, we used a simulation as shown in Table 4.

[Lines 414, 592, 724: It is Sjögren or Sjögren’s syndrome]

We have corrected “Sjogren” and “Sjögren’s” to “Sjögren” (Revised manuscript, Table 1 and line 540).

---

## [Editor Report · Decision Letter 2]

10 Jun 2020

Mass spectrometry–based absolute quantification of amyloid proteins in pathology tissue specimens: merits and limitations

PONE-D-20-02311R2

Dear Dr. Fukayama,

We’re pleased to inform you that your manuscript has been judged scientifically suitable for publication and will be formally accepted for publication once it meets all outstanding technical requirements.

Kind regards,

Harald Mischak

Academic Editor

PLOS ONE
---

## [Editor Report · Acceptance letter]

19 Jun 2020

PONE-D-20-02311R2 

Mass spectrometry-based absolute quantification of amyloid proteins in pathology tissue specimens: merits and limitations 

Dear Dr. Fukayama:

I'm pleased to inform you that your manuscript has been deemed suitable for publication in PLOS ONE. Congratulations! Your manuscript is now with our production department. 

Kind regards, 

on behalf of

Prof. Harald Mischak 

Academic Editor

PLOS ONE